# Istaroxime treatment ameliorates calcium dysregulation in a zebrafish model of phospholamban R14del cardiomyopathy

S. M. Kamel [1,7], C. J. M. van Opbergen [2,7], C. D. Koopman[1,2,7], A. O. Verkerk [3,4], B. J. D. Boukens[3,4], B. de Jonge[3], Y. L. Onderwater [1], E. van Alebeek[1], S. Chocron[1], C. Polidoro Pontalti[2], W. J. Weuring [5], M. A. Vos[2], T. P. de Boer [2], T. A. B. van Veen [2,8✉] & J. Bakkers [1,2,6,8✉]

The heterozygous *Phospholamban* p.Arg14del mutation is found in patients with dilated or arrhythmogenic cardiomyopathy. This mutation triggers cardiac contractile dysfunction and arrhythmogenesis by affecting intracellular $Ca^{2+}$ dynamics. Little is known about the physiological processes preceding induced cardiomyopathy, which is characterized by subepicardial accumulation of fibrofatty tissue, and a specific drug treatment is currently lacking. Here, we address these issues using a knock-in *Phospholamban* p.Arg14del zebrafish model. Hearts from adult zebrafish with this mutation display age-related remodeling with subepicardial inflammation and fibrosis. Echocardiography reveals contractile variations before overt structural changes occur, which correlates at the cellular level with action potential duration alternans. These functional alterations are preceded by diminished $Ca^{2+}$ transient amplitudes in embryonic hearts as well as an increase in diastolic $Ca^{2+}$ level, slower $Ca^{2+}$ transient decay and longer $Ca^{2+}$ transients in cells of adult hearts. We find that istaroxime treatment ameliorates the in vivo $Ca^{2+}$ dysregulation, rescues the cellular action potential duration alternans, while it improves cardiac relaxation. Thus, we present insight into the pathophysiology of *Phospholamban* p.Arg14del cardiomyopathy.

[1] Hubrecht Institute, Royal Netherlands Academy of Arts and Sciences (KNAW), University Medical Centre Utrecht, 3584 CT Utrecht, The Netherlands. [2] Department of Medical Physiology, Division of Heart & Lungs, University Medical Center Utrecht, Yalelaan 50, 3584 CM Utrecht, The Netherlands. [3] Department of Medical Biology, Amsterdam Cardiovascular Sciences, University of Amsterdam, Amsterdam University Medical Centers, Amsterdam, The Netherlands. [4] Department of Experimental Cardiology, University of Amsterdam, Amsterdam University Medical Centers, Amsterdam, The Netherlands. [5] Department of Genetics, UMC Utrecht Brain Center, University Medical Center Utrecht, Utrecht University, Utrecht, The Netherlands. [6] Department of Pediatric Cardiology, Division of Pediatrics, University Medical Center Utrecht, Utrecht, The Netherlands. [7]These authors contributed equally: S. M. Kamel, C. J. M. van Opbergen, C. D. Koopman. [8]These authors jointly supervised this work: van Veen T.A.B., J. Bakkers. ✉email: A.A.B.vanVeen@umcutrecht.nl; j.bakkers@hubrecht.eu

Arrhythmogenic cardiomyopathy (ACM) is a genetically inherited disease that occurs in 1:1000–1:5000 of the general population. ACM is characterized by the accumulation of fibrous tissue and fat at the sub-epicardial region of the ventricles, where it infiltrates the myocardial wall. In addition, malignant ventricular arrhythmias and a high propensity for sudden cardiac death have been observed[1]. Sudden cardiac death most commonly occurs in young patients, during the concealed stage of the disease, when no overt cardiomyopathy is detectable yet. Little is known about the underlying cause of cardiac remodeling, eventually culminating in contractile dysfunction, heart failure, and sudden cardiac death in ACM patients[2]. Myocardial biopsies of late-stage ACM patients also display inflammatory infiltrates, which are associated with the fibrofatty replacement of myocardial tissue[2]. Most cases of familial ACM are caused by mutations in desmosomal genes (e.g. *plakophilin*, *desmoplakin*). However, non-desmosomal gene mutations such as the *Phospholamban* (*PLN*) p.Arg-14 deletion (R14del) have recently been identified as a cause for ACM as well[3].

*PLN* R14del is a Dutch founder mutation and the most prevalent cardiomyopathy-related mutation in the Netherlands. It has been identified in 10–15% of all Dutch patients with dilated cardiomyopathy and/or ACM and it is estimated that there are more than 1000 *PLN*-R14del Dutch carriers[4]. *PLN* mutations, including *PLN* R14del, have also been detected in several other countries, including Canada, the USA, Spain, Germany, and Greece[5–8]. Patient phenotypes are very heterogeneous not only between families but also within families[9,10]. Up to today, no homozygous carriers have been identified[11]. Severely affected *PLN* R14del mutation carriers display profound fibrofatty infiltration in the sub-epicardial region of both ventricular walls, along with malignant ventricular arrhythmias and end-stage heart failure[12]. Currently, specific drugs to treat this disease are lacking.

PLN is a small 52 amino-acid transmembrane sarcoplasmic reticulum (SR) protein and a crucial regulator of SR function within cardiomyocytes[13]. During excitation–contraction coupling, free cytosolic $Ca^{2+}$ levels increase, causing more $Ca^{2+}$ to bind to the cardiac myofilaments, which generates contractile force. After contraction, free cytosolic $Ca^{2+}$ levels must be diminished to resting levels in order to induce cardiomyocyte relaxation. $Ca^{2+}$ is pumped back into the SR via the sarco(endo) plasmic reticulum $Ca^{2+}$ ATPase (SERCA2a) and extruded out of the cell via the $Na^+$–$Ca^{2+}$ exchanger (NCX). Activity of SERCA2a, the cardiac SERCA isoform, is tightly regulated by its scaffolding protein PLN. Under physiological conditions, PLN inhibits SERCA2a activity and thereby tempers the rate at which $Ca^{2+}$ flows back into the SR. Upon the inhibition of PLN activity, such as by phosphorylation on $Ser^{16}$, SR $Ca^{2+}$ cycling is enhanced, leading to an increase in SR $Ca^{2+}$ reuptake and improved cardiac function[14–16]. The *PLN* R14del mutation results in a protein that has a stronger affinity for SERCA2a[6], which has been correlated with the disruption of the $Ser^{16}$ phosphorylation motif. It was suggested that disruption of this motif could lead to an increased inhibition of the SERCA2a, cytoplasmic $Ca^{2+}$ overload, and increased risk for malignant ventricular arrhythmias[17,18]. Whether changes in intracellular $Ca^{2+}$ dynamics also occur in vivo and affect cardiac structure and function remains unaddressed. Studying these processes in depth requires accurate in vivo models. A recent knock-in mouse model demonstrated severe cardiac remodeling and heart failure when homozygous for *PLN* R14del. Heterozygous *PLN* R41del mice only showed mild fibrosis, whereas an inflammatory response or heart failure was never observed[19].

During the past decades, the zebrafish (*Danio rerio*) has emerged as a powerful, cost-efficient, and easy-to-use vertebrate model to study human disease[20]. Its conserved genome (82% of all human disease-causing genes has at least one zebrafish orthologue), physiology, and pharmacology has made the zebrafish a highly valuable model for the resolution of human disease-related mechanisms and for novel drug (target) discovery[21–23]. In addition, CRISPR/Cas genome editing in the zebrafish has led to the opportunity of introducing of patient-specific gene mutations in the zebrafish genome[24–26].

In this study, we used zebrafish with a R14del variant in the endogenous *plna* gene to understand the cardiac pathophysiology caused by the *PLN* R14del mutation. Strikingly, an age-related severe cardiac remodeling was observed at the sub-epicardial region of the heart, combined with a strong infiltration of immune cells, fibroblasts, and fat deposits. In young-adult *plna* R14del zebrafish without apparent structural remodeling, we observed variations in ventricular outflow peak velocity between consecutive beats. In correlation with this, irregular action potential (AP) duration (APD) (alternans) was identified in isolated *plna* R14del cardiomyocytes. Since these irregular APs can be caused by altered $Ca^{2+}$ dynamics, we analyzed intracellular $Ca^{2+}$ dynamics in vivo. Corroborating this, we found evidence for delayed SR $Ca^{2+}$ reuptake in *plna* R14del cardiomyocytes. Importantly, istaroxime ameliorated the in vivo $Ca^{2+}$ dysregulation and contractile impairment in *plna* R14del zebrafish, and rescued the observed irregular APDs in *plna* R14del cardiomyocytes.

## Results

**Cardiac remodeling in adult *plna* R14del zebrafish.** A zebrafish homolog of human *PLN* has been annotate on chromosome 20 (ENSDARG00000069404), which we refer to as *plna* (Fig. S1A). Subsequent screening for homologs sequences within the zebrafish genome revealed a second *pln*-like gene on chromosome 17 (si:ch211–270g19.5; ENSDARG00000097256), which we refer to as *plnb* (Fig. S1A). Both zebrafish genes are expressed in the heart and have similar expression levels in the ventricle, while in the atrium *plnb* is most abundant (Fig. S1C). Both genes contain a short predicted open reading frame with high sequence similarity compared to human PLN (75% for Plna and 67% for Plnb) (Fig. S1B). To get a better understanding of the consequences of the *PLN* R14del mutation, we engineered a fish line with a 3 bp in-frame deletion in *plna* removing the conserved arginine at positions 14, which we will refer to as *plna* R14del (Figs. S1D and S2)[25]. Since the expression levels in the ventricle for *plna* and *plnb* are similar, we reasoned that zebrafish with a homozygous *plna* R14del mutation and wild type for the *plnb* gene (*plna* R14del/ R14del; *plnb* +/+) would be the best genotype to represent the heterozygous PLN R14del mutation in patients. We will refer to this genotype as homozygous *plna* R14del fish or simply as *plna* R14del mutants. Adult fish with either a heterozygous or homozygous *plna* R14del mutation had a normal appearance and showed normal behavior. To investigate cardiac morphological changes, we isolated hearts from 2-year-old wild-type control, *plna* R14del heterozygous, and homozygous zebrafish. Importantly, we observed that 23% of homozygous *plna* R14del fish displayed severe cardiac morphological changes, such as an increased size of the heart and a white flocculent layer of tissue that lined the outside of the heart. We did not observe any of these changes in wild-type siblings nor in heterozygous *plna* R14del fish (Fig. 1A–C). Histological analysis of the remodeled hearts revealed altered tissue organization in the sub-epicardial region with areas of high nuclear density and areas that appeared acellular compared to wild-type siblings and non-remodeled hearts (Fig. 1D–F). Moreover, Picro-sirius red staining revealed collagen deposition in the sub-epicardial region of these remodeled hearts, unlike wild-type siblings and non-remodeled hearts (Fig. 1G–I). To address whether the morphological changes

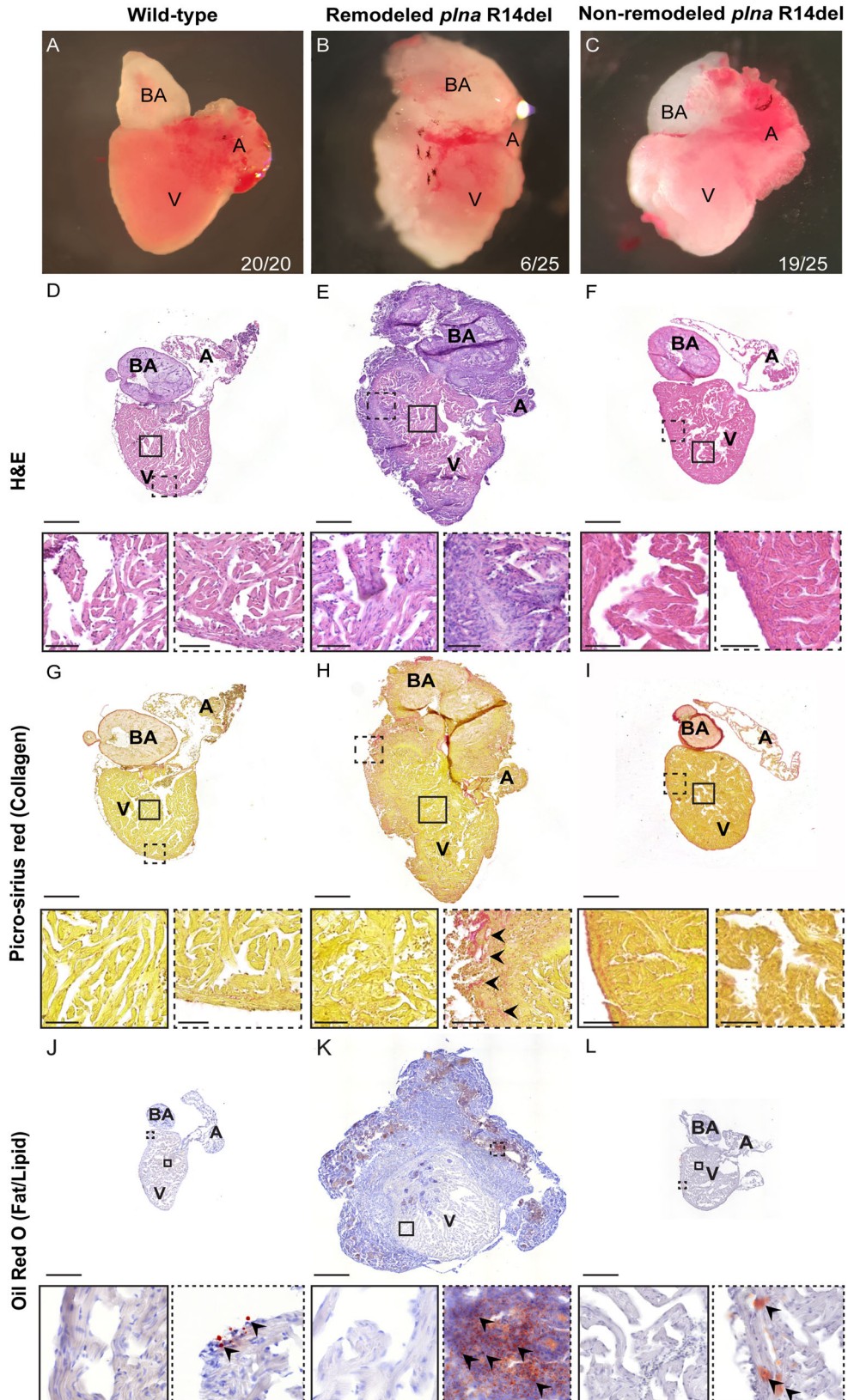

**Fig. 1 Structural remodeling of the adult *plna* R14del zebrafish heart. A–C** Bright-field images of isolated adult zebrafish hearts, 2 years of age: wild-type fish, remodeled *plna* R14del mutant heart and non-remodeled *plna* R14del heart. **D–F** Hematoxylin and eosin staining of the three conditions to identify nuclei, with zoom-in at indicated regions. **G–I** Picro-sirius red staining of collagen deposition for the three conditions, collagen fibers are shown as red staining. **J–L** Oil Red O staining for fat/lipid of the three conditions. All stainings were performed on WT $n = 3$, *plna* R14del $n = 3$, two experimental replicates. Zoom-in of each indicated region is included. Images were taken at a magnification of ×20. Scale bars are 200 µm for whole-heart tile scans and 50 µm for zoom-in regions. A atrium, V ventricle, BA bulbus arteriosus.

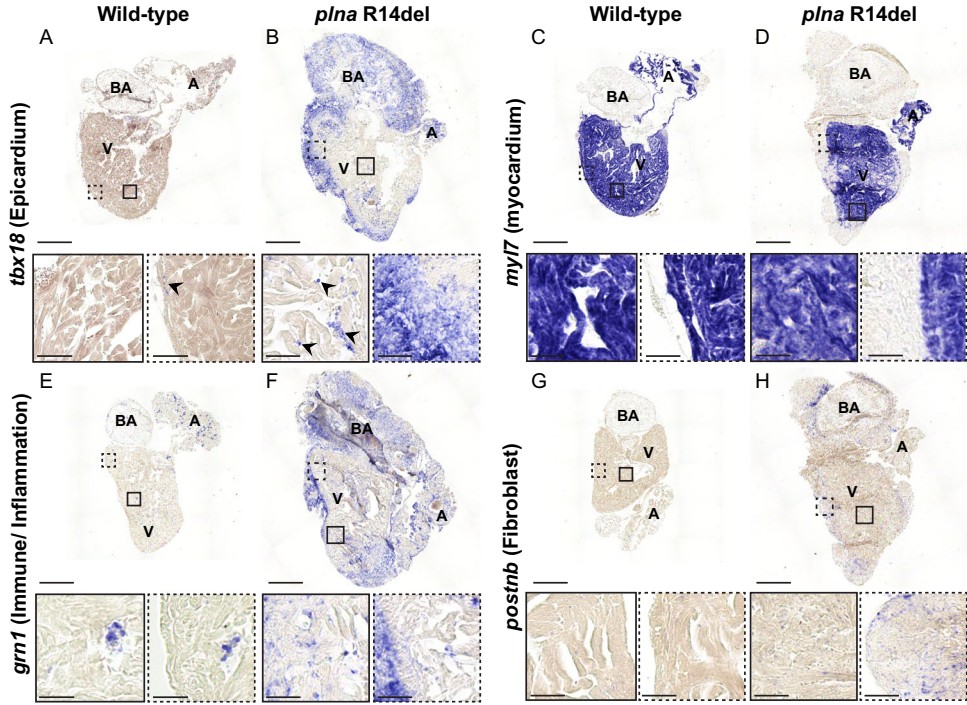

**Fig. 2 Epicardial response and immune infiltration in adult hearts of *plna* R14del zebrafish.** In situ hybridization on wild type and *plna* R14del mutant adult zebrafish hearts, 2 years of age. **A**, **B** Expression of *tbx18* to indicate the epicardial cells. **C**, **D** Expression of *myl7* to indicate the myocardial cells. **E**, **F** Expression of *grn1* to indicate the immune cells. **G**, **H** Expression of *postnb* to indicate fibroblasts. All stainings were performed on WT $n = 3$, *plna* R14del $n = 3$, two experimental replicates. Zoom-in of each region is indicated. Images were taken at a magnification of ×20. Scale bars are 200 μm for whole-heart tile scans and 50 μm for zoom-in regions. A atrium, V ventricle, BA bulbus arteriosus.

already occur earlier, we isolated hearts from 10-month homozygous *plna* R14del fish ($n = 13$) and observed one case with a similar altered tissue organization, which was accompanied by cell death, in the sub-epicardial region (Fig. S3). Interestingly, Oil Red O staining revealed prominent fat deposits throughout the sub-epicardial tissue and especially in the acellular areas (Fig. 1J–L). To test whether the presence of wild-type Plnb could explain the low penetrance of the cardiac remodeling, we mutated the *plnb* gene by CRISPR/Cas9-mediated gene editing. We identified F1 fish with a 5 bp deletion in *plnb*, which is expected to result in a frameshift and truncation of the Plnb protein (Fig. S1E). Adult fish homozygous for the *plnb*-truncating mutation and homozygous for the *plna* R14del mutation showed earlier and more severe cardiac remodeling and fat deposits with higher penetrance (Fig. S4). Overall, these results suggest that zebrafish with a homozygous R14del mutation in *plna* undergo age-related cardiac remodeling most severe in the sub-epicardial region. The severity and penetrance of the phenotype depend on the level of wild-type Plnb.

**Epicardial responses and immune infiltration in adult hearts of *plna* R14del zebrafish.** Next, we identified the cell types involved in the observed cardiac remodeling by investigating the presence of cell-type-specific markers using in situ hybridization (ISH). T-box18 (Tbx18) is a transcription factor that is weakly expressed in the epicardium of the adult heart, but its expression is induced upon cardiac injury[27]. Indeed, in wild-type hearts, we found expression of *tbx18* to be restricted to sparse cells in the sub-epicardial region (Fig. 2A). Strikingly, expression of *tbx18* was clearly present in the epicardial region of *plna* R14del mutants, with *tbx18*-expressing cells forming several cell layers that covered the ventricle and the bulbus arteriosus (arrows in Fig. 2B)

and not overlapping with the myocardium, marked by *myl7* (Fig. 2C, D). In addition, dispersed *tbx18*-positive cells intermingled with cardiomyocytes were present in the ventricle of remodeled *plna* R14del mutant hearts, something we never observed in the hearts of wild types (Fig. 2B; boxes). Since TUNEL-staining revealed apoptotic cells in the sub-epicardial region we analyzed the presence of immune cells by granulin (*grn1*) expression. Corroborating this finding, an accumulation of immune cells in the sub-epicardial region was observed as well (Fig. 2E, F). As Picro-sirius red staining indicated enhanced fibrosis in the *plna* R14del mutant hearts, we analyzed the presence of fibroblasts, marked by the expression of periostin (*postnb*). Consistent with the increased fibrosis, we observed periostin expression in cells located in the sub-epicardial region (Fig. 2G, H). Together these results indicate that cardiac remodeling of the sub-epicardial region in *plna* R14del mutants is accompanied by cellular damage, an inflammatory response and fibrosis.

**Cardiac contractility defects in the *plna* R14del mutant heart.** Structural remodeling of the heart is often preceded by impaired cardiac contractile dysfunction[28]. To examine cardiac contractility and pump function in *plna* R14del mutant zebrafish, and relate this to heart morphology, we performed echocardiographic measurements combined with histological analysis of the cardiac tissue. For echocardiography fish were positioned ventral side up and a long-axis view, which included the two chambers of the heart, was imaged accordingly (Fig. 3A, B). To track changes in cardiac performance over time, echocardiography of fish at two different ages (6 and 10 months old) was performed. Color Doppler images were used to examine contractile parameters. While the heart rates were similar between groups, cardiac output

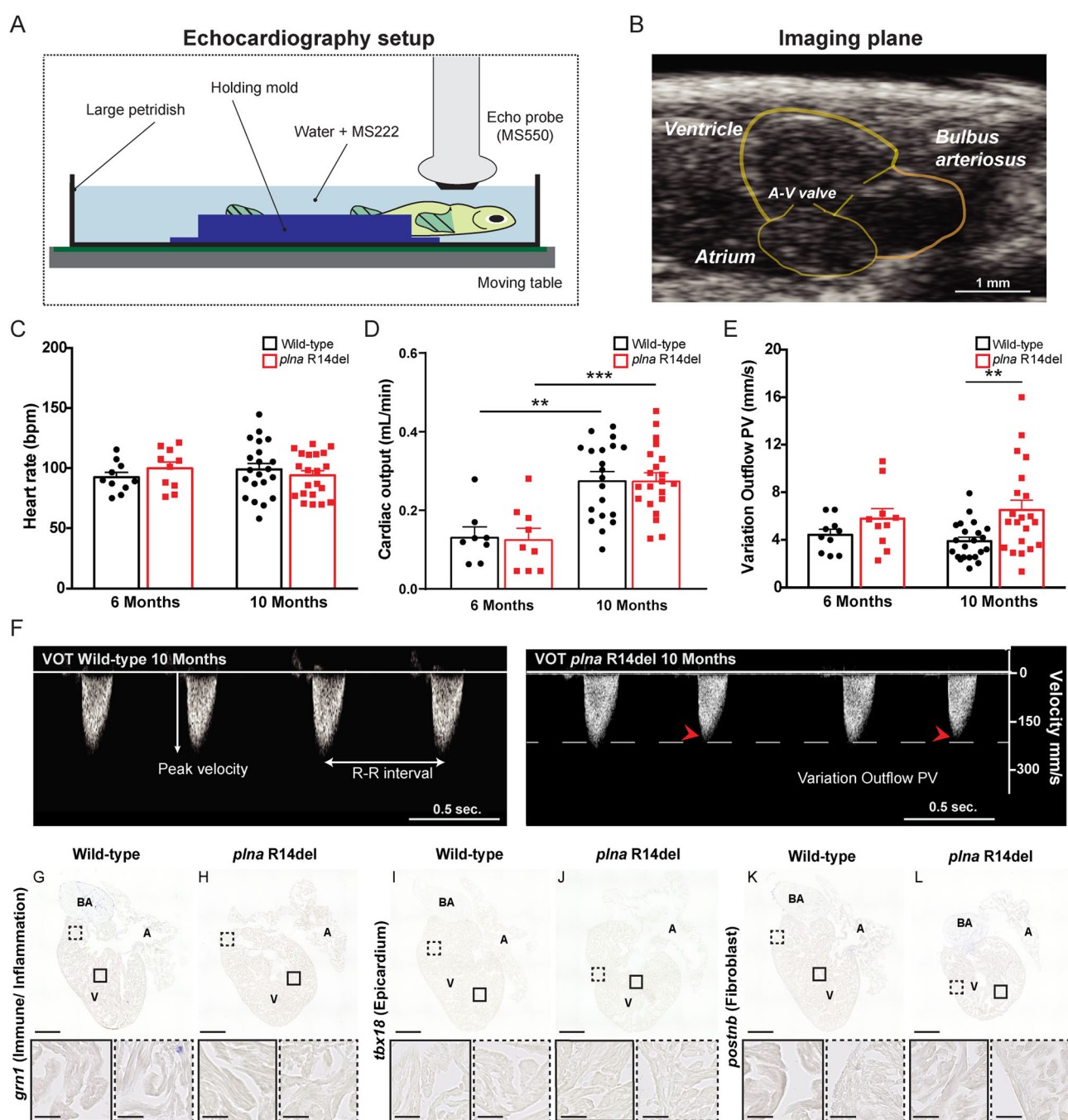

**Fig. 3 Contractile variations in the structural normal adult *plna* R14del zebrafish heart. A** Graphical illustration of the zebrafish echocardiography imaging setup. **B** B-mode echocardiography imaging plane of the adult zebrafish heart, ventricular walls, and cardiac valves are depicted in yellow. **C** Heart rate of wild type (WT) and *plna* R14del mutant zebrafish, 6 and 10 months of age (mean ± SEM, WT *n* = 10, *plna* R14del *n* = 21, two-way ANOVA). **D** Cardiac output of WT and *plna* R14del zebrafish, 6 and 10 months of age (mean ± SEM, **p = 0.0013, ***p = 0.004, WT *n* = 10, *plna* R14del *n* = 21, one-way ANOVA). **E** Variation in outflow peak velocity in WT and *plna* R14del zebrafish, 6 and 10 months of age (mean ± SEM, **p = 0.01954, WT *n* = 10, *plna* R14del *n* = 21, one-way ANOVA). **F** Representative examples of color Doppler ventricular outflow measurements in 10-month-old WT and *plna* R14del zebrafish. Dashed white and red arrows indicate maximum and minimum peak velocities, respectively. All measurements were performed in two experimental replicates. **G**–**L** Representative images of in situ hybridization of these hearts with markers to observe immune/inflammation, epicardium, and fibroblasts (using *grn1*, *tbx18*, and *postnb*, respectively). All stainings were performed on WT *n* = 3, *plna* R14del *n* = 3, two experimental replicates. Images were taken at a magnification of ×20. Scale bars are 200 μm for whole-heart tile scans and 50 μm for zoom-in regions. Wild type are highlighted in black and *plna* R14del in red. A atrium, V ventricle, BA bulbus arteriosus, A–V valve atrial–ventricular valve, Vent. outflow ventricular outflow, variation outflow PV variation outflow peak velocity, bpm beats per minute, ml/min milliliter per minute, mm/s millimeter per second, VOT ventricular outflow tract. Source data are provided as a Source Data file.

was significantly higher in 10-month-old wild type and *plna* R14del mutant fish, compared to 6-month-old fish (Fig. 3C, D and Table S2). Interestingly, *plna* R14del mutant fish showed variations in ventricular outflow peak velocity (VOT PV) between beats. These variations were already observed in 6-month-old *plna* R14del mutant fish, and became more apparent in 10-month-old fish, resulting in a significantly higher extent of VOT PV variation in *plna* R14del mutant fish, compared to their wild-type siblings (Fig. 3E, F). Histological analysis of hearts with strong VOT PV variations revealed no accumulation of immune cells or fibroblast and no signs of cardiac remodeling (Fig. 3G–L). Taken together, these results indicate that the *plna* R14del mutation causes beat-to-beat variations in cardiac output in the absence of cardiac remodeling. This suggests that the cardiac contractile dysfunction is not caused by cardiac remodeling, but may be causal for cardiac remodeling.

**Cellular electrophysiology of *plna* R14del ventricular cardiomyocytes.** Variations in outflow peak velocity can be the consequence of aberrancies in cardiac cellular electrophysiology and/or $Ca^{2+}$ dynamics (reviewed by Edwards and Blatter[29] and Kulkarni et al.[30]). To examine cardiac electrophysiology in *plna* R14del mutant zebrafish we first recorded electrocardiograms (ECGs) from 10-month-old fish. The ECGs revealed normal PR- and PQ interval times and normal amplitudes of the P- and R-waves (Fig. S5), indicating that electrical conduction from the atrium to the ventricle was unaffected in *plna* R14del carriers at baseline conditions. Next, we addressed whether *plna* R14del affects APs and membrane currents by performing patch clamp analysis on isolated cardiomyocytes from 10-month-old fish. APs and membrane currents were measured at a steady-state condition using the perforated patch clamp technique to limit technical disturbances in intracellular $Ca^{2+}$ homeostasis. None of the AP parameters measured at 1 Hz differed significantly between WT and *plna* R14del mutant APs (Fig. 4A, B). With an increase of the stimulus frequency, however, we observed a remarkable difference between both groups. While the average $APD_{90}$-frequency relationships are virtually overlapping (Fig. 4C), we noticed clear alternations in APD between consecutive APs (electrical alternans) in *plna* R14del mutant cells at higher pacing frequencies. Figure 4D shows a typical example of such an alternans. On average, the alternans in AP between consecutive beats was approximately four times larger in *plna* R14del mutant cells compared to cardiomyocytes from their wild-type siblings (Fig. 4E).

We next measured the major membrane currents responsible for AP repolarization in zebrafish cardiomyocytes[30]. $I_{K1}$, defined as steady-state current negative of $-30$ mV (Fig. S6A), was not significantly different between WT and *plna* R14del mutant cardiomyocytes. This finding is in line with the comparable resting membrane potential (RMP) and Phase-3 repolarization velocities between the two groups (Fig. 4B). In addition, $I_{Kr}$ defined as steady-state current positive of $-30$ mV (Fig. S6A) was unaffected in *plna* R14del mutant cardiomyocytes. Finally, we analyzed $I_{Ca,L}$ density and L-type $Ca^{2+}$-channel (LTCC) gating properties. Current density was not affected by the *plna* R14del mutation (Fig. S6B). Together these data indicate that the *plna* R14del mutation does not affect the main ion channels underlying AP morphology.

**Impaired $Ca^{2+}$ dynamics in *plna* R14del mutant hearts.** AP alternans are frequently caused by a dysfunction of intracellular $Ca^{2+}$ homeostasis and may trigger cardiac arrhythmias[31,32]. To study the effects of the *plna* R14del mutation on $Ca^{2+}$ dynamics,

we loaded the isolated adult cardiomyocytes with the ratiometric $Ca^{2+}$ dye Indo-1 and calculated intracellular $[Ca^{2+}]$. We observed an increased diastolic $[Ca^{2+}]_i$, longer $Ca^{2+}$ transients (time to 20 and 50% recovery), and slower decay of the $Ca^{2+}$ transient, which is consistent with a slower SR $Ca^{2+}$ reuptake (Fig. 5A–F). To exclude that these changes were due to cardiac remodeling that takes place in the adult *plna* R14del fish we also measured $Ca^{2+}$ dynamics in the embryonic heart. We generated transgenic *plna* R14del zebrafish, expressing a cytosolic cardiac $Ca^{2+}$ sensor *tg(myl7:Gal4FF UAS:GCaMP6f)*. Analysis of GCaMP6f (cpEGFP) signal intensity over time allowed examination of $Ca^{2+}$ transient amplitudes, diastolic $Ca^{2+}$ levels, and the speed of intracellular $Ca^{2+}$ release and reuptake/clearance, as described earlier (Fig. 5G)[33]. Here, we used 3-day-old embryos since they allow in vivo analysis of intracellular $Ca^{2+}$ dynamics due to their optical transparency. No significant difference was observed in $Ca^{2+}$ transient frequency (Fig. 5H). However, the $Ca^{2+}$ transient amplitude was on average 26% lower in *plna* R14del mutant embryos compared to wild-type siblings (WT: $99.8 \pm 6.12\%$ and *plna*-R14del: $73.9 \pm 9.03$, $p \leq 0.05$) (Fig. 5J). A similar decrease in $Ca^{2+}$ transient amplitude was also observed when wild-type embryos were treated with the SERCA inhibitor thapsigargin (Fig. S7), suggesting that SR $Ca^{2+}$ sequestration is impaired in the presence of the *plna* R14del mutation. In addition, we observed a trend towards prolonged $Ca^{2+}$ transient recovery time (WT: $123.84 \pm 7.28$ ms and *plna*-R14del: $130.89 \pm 8.57$ ms, $p = $ n.s.) and significantly slower decay of the $Ca^{2+}$ transient, as indicated by an increased decay time constant ($\tau$ of mono-exponential fits of WT: $105.74 \pm 14.35$ ms and *plna*-R14del: $178.6 \pm 30.78$ ms, $p \leq 0.05$) between wild-type siblings and *plna* R14del mutants (Fig. 5I, L). Diastolic $Ca^{2+}$ levels were not significantly affected in *pln*a R14del mutant embryos (Fig. 5K). Together these results demonstrate that intracellular $Ca^{2+}$ dynamics in cardiomyocytes of the embryonic and adult heart are affected by the *plna* R14del mutation and suggest that *plna* R14del impairs SR $Ca^{2+}$ reuptake.

**Istaroxime enhances cardiac output by improving ventricular relaxation via targeting PLN-SERCA interaction.** Since a specific drug treatment is lacking to treat patients with *PLN* R14Del cardiomyopathy, we looked for a small molecule with the potential to rescue the above-described electrophysiological phenotypes observed in *plna* R14del mutants. Istaroxime is proposed to stimulate the activity of SERCA2a, thereby enhancing SR $Ca^{2+}$ sequestration[34]. In addition, it inhibits $Na^+/K^+$ ATPase (NKA) activity[35,36]. Since our results indicate that SR $Ca^{2+}$ sequestration is impaired in *plna* R14del mutant zebrafish, we tested whether the treatment of *plna* R14del mutant embryos with istaroxime could rescue the $Ca^{2+}$ abnormalities. Wild type and *plna* R14del mutant embryos with GCaMP6f were first imaged to record baseline $Ca^{2+}$ dynamics. After baseline recording, 100 μM istaroxime was added to the E3-MS222 medium and after 30 min of incubation another recording was performed (Fig. 6A). One hundred micromolar istaroxime was determined as the optimal drug concentration in a dose–response experiment using wild-type GCaMP6f fish (Fig. S8). When combining the recordings of the GCaMP6f intensities from individual embryos before and after istaroxime treatment, we observed a restoration of the $Ca^{2+}$ transient amplitude and decay of the $Ca^{2+}$ transient in *plna* R14del mutants to wild-type levels (Fig. 6B, C, E), leaving diastolic $Ca^{2+}$ levels unaffected (Fig. 6D).

Next, we examined whether the effect of istaroxime on $Ca^{2+}$ transient amplitude also affected cardiac contractility using high-speed video imaging. As expected, istaroxime significantly

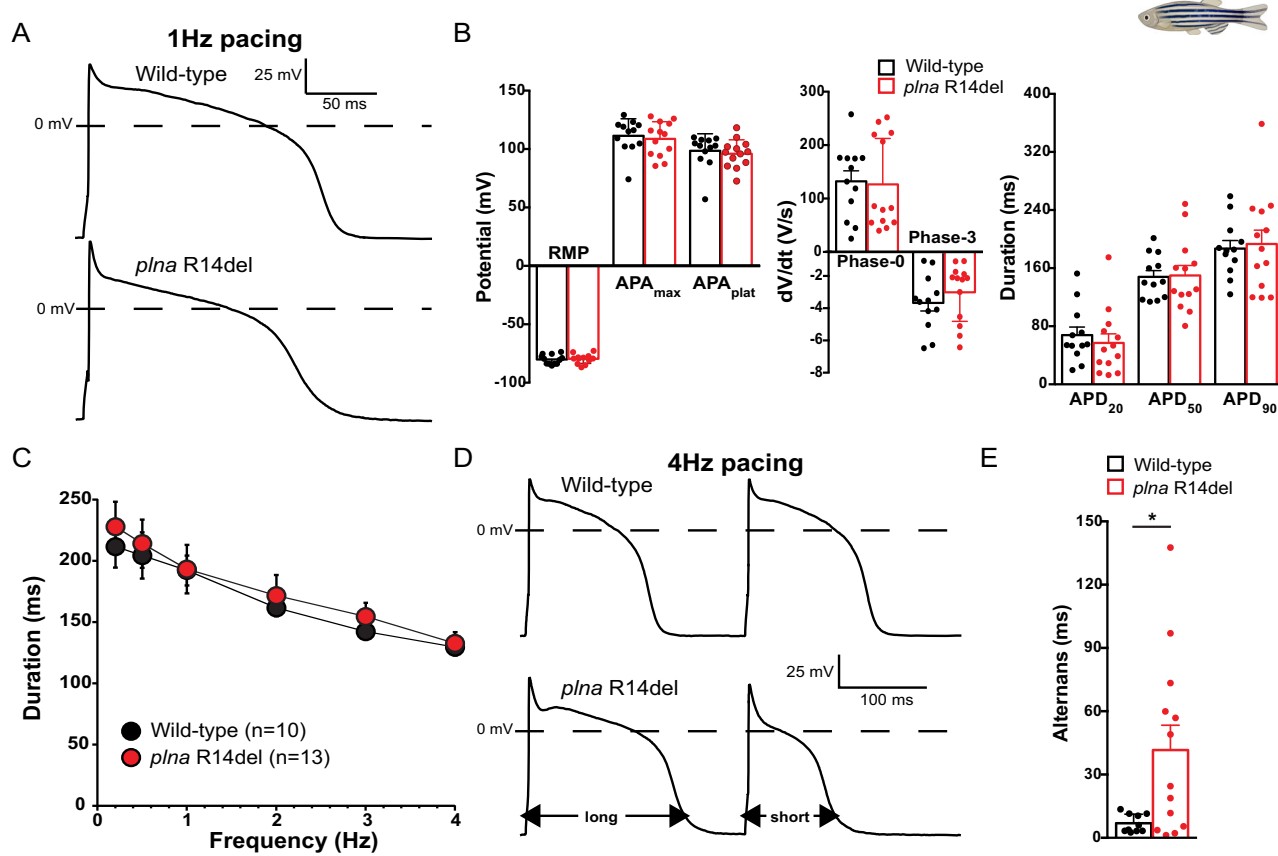

**Fig. 4 Action potential alternans in *plna* R14del isolated cardiomyocytes. A** Typical action potentials (APs) of cardiomyocytes isolated from a 10-month-old wild type (WT) (top panel) and *plna* R14del mutant (bottom panel) zebrafish, paced at 1 Hz. **B** Average AP parameters at 1 Hz pacing (mean ± SEM, WT $n = 12$, *plna* R14del $n = 13$, unpaired Students *t*-test). **C** AP duration at 90% of repolarization (APD$_{90}$) at 0.2–4 Hz pacing (mean ± SEM, WT $n = 10$, *plna* R14del $n = 13$, unpaired Students *t*-test). **D** Two consecutive APs during 4 Hz pacing in a WT (top panel) and *plna* R14del (bottom panel) cardiomyocyte. The *plna* R14del cardiomyocyte showed an AP alternans where a long AP is followed by a short AP. **E** Average APD$_{90}$ difference between two consecutive APs (alternans) during 4 Hz pacing (mean ± SEM, *$p = 0.044$, WT $n = 10$, *plna* R14del $n = 13$, Mann–Whitney rank-sum test). Wild type are highlighted in black and *plna* R14del in red. WT wild type, Hz Hertz, mV millivolt, ms milliseconds, V/s Volts per second, RMP resting membrane potential, APA$_{max}$ maximal action potential amplitude, APA$_{plat}$ action potential amplitude at plateau phase, APD$_{20}$ AP duration at 20% of repolarization, APD$_{50}$ AP duration at 50% of repolarization, APD$_{90}$ AP duration at 90% of repolarization. Source data are provided as a Source Data file.

enhanced stroke volume, cardiac output, and ejection fraction in both wild type and *plna* R14del mutants (Figs. 6F and S9), which was not observed with a placebo treatment (Fig. S10). Stroke volume and cardiac output were likely affected via an improvement of the diastolic filling of the heart, as only end-diastolic, and not end-systolic volume, was changed by istaroxime treatment (Fig. 6G, H). In order to further elucidate the effect of istaroxime on the rate of myocardial relaxation (lusitropy), we examined contractile cycle length. The total duration of contraction was divided into contraction time (time from maximal relaxation to maximal contraction) and relaxation time (time from maximal contraction to maximal relaxation). Total contraction duration and contraction time were not different between *plna* R14del mutants and wild-type embryos and were not affected by istaroxime treatment (Figs. 6I and S9). Interestingly, the relaxation time was slightly longer in *plna* R14del mutants at baseline and significantly decreased after istaroxime treatment (Fig. 6J).

Since istaroxime is known to block NKA and potentiate SERCA2a activity, we tested whether istaroxime improves the $Ca^{2+}$ transient amplitude via SERCA2a activation or NKA inhibition. Therefore, we compared the effect of istaroxime and

ouabain, a specific NKA blocker, on ventricular $Ca^{2+}$ transient dynamics in wild-type embryos. In contrast to istaroxime, ouabain treatment had no effect on $Ca^{2+}$ transient amplitude (Fig. S11A, B). To test whether a NKA block could be responsible for the lusitropic effects of istaroxime, potential changes in contractility were examined in ouabain-treated embryos. Ouabain did not significantly enhance stroke volume, ejection fraction, or cardiac output (Fig. S11C–F). Next, we validated the effectiveness of ouabain in our zebrafish model using embryos expressing a genetically encoded Voltage Sensitive Fluorescent Protein (VSFP Butterfly-CY) (Fig. S12A). Fluorescent high-speed recordings of VSFP embryos demonstrated a clear APD shortening in the presence of ouabain compared to baseline values (Fig. S12B).

Together, these data indicate that istaroxime can improve SR $Ca^{2+}$ reuptake in *plna* R14del mutant cardiomyocytes, an effect that is predominantly asserted via interference of the PLN–SERCA interaction.

**Istaroxime shortens repolarization and suppresses APD alternans *in plna* R14del cardiomyocytes.** To test whether istaroxime could also rescue the AP alternans that we observed in

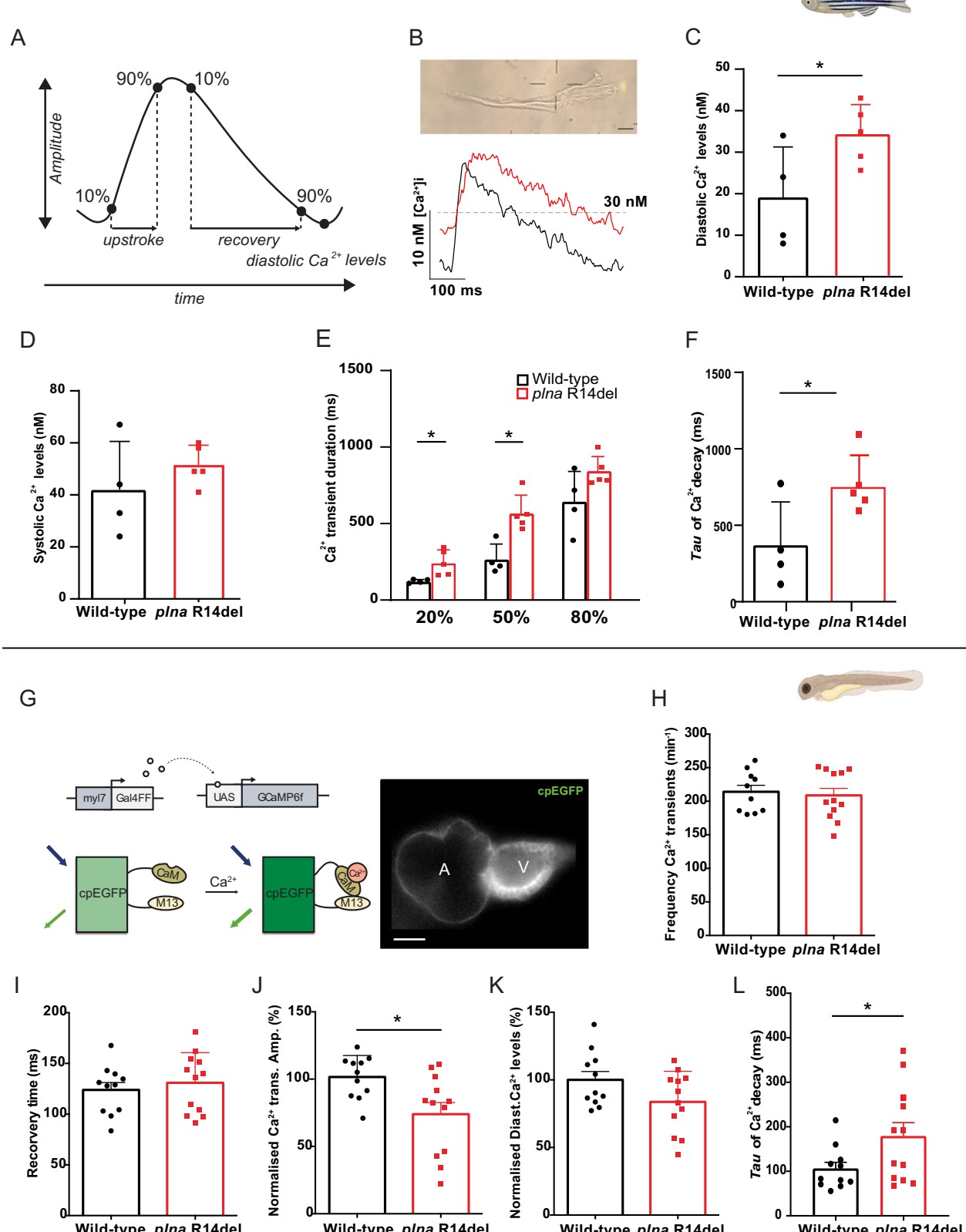

adult *plna* R14del cardiomyocytes, we performed patch clamp recordings on isolated ventricular cardiomyocytes. To this end, cardiomyocytes of adult wild type and *plna* R14del zebrafish hearts were measured at 1 and 4 Hz before and after treatment with istaroxime. Interestingly, istaroxime treatment resulted in

a significant shortening of $APD_{20}$, $APD_{50}$, and $APD_{90}$ and a complete absence of APD alternans (Fig. 7A–D). The effects of istaroxime on APD were confirmed in vivo, using high-speed fluorescence imaging on embryos expressing VSFP Butterfly-CY (Fig. S12C).

**Fig. 5 Calcium dynamic changes in adult ventricular cardiomyocytes and embryonic plna R14del zebrafish. A** Schematic representation of $Ca^{2+}$ transient parameters showing $Ca^{2+}$ transient upstroke time, $Ca^{2+}$ transient recovery time, $Ca^{2+}$ transient amplitude, and diastolic $Ca^{2+}$ levels. **B** Representative image of an isolated ventricular cardiomyocyte and a $Ca^{2+}$ trace from adult ventricular cardiomyocyte of wild type and plna R14del mutants. Intracellular $Ca^{2+}$ parameters were measured in adult ventricular cardiomyocytes of wild type (WT) and plna R14del, including **C** diastolic $Ca^{2+}$ levels (mean ± SEM, *$p = 0.05$, WT $n = 4$, plna R14del $n = 5$), **D** systolic $Ca^{2+}$ levels, **E** Calcium transient duration at 20%, 50, and 80% (mean ± SEM, CaD20: $p = 0.026$ and CaD50: $p = 0.005$, WT $n = 4$, plna R14del $n = 5$), and **F** Tau of $Ca^{2+}$ decay (mean ± SEM, *$p = 0.041$, WT $n = 4$, plna R14del $n = 5$). All measurements were performed in three experimental replicates. **G** DNA construct and sensor dynamics of GCaMP6f (left panel). GCaMP6f was placed under the control of the myl7 promoter to restrict its expression to the heart. The Gal4FF-UAS system amplifies its expression. GCaMP6f consists of a circularly permutated enhanced green fluorescence protein (cpEGFP) fused to calmodulin (CaM) and the M13 peptide. When intracellular calcium ($Ca^{2+}$) rises, CaM binds to M13, causing increased brightness of cpEGFP. Using a high-speed epifluorescence microscope, movies of 3 dpf non-contracting GCaMP6f embryonic hearts were recorded. Several intracellular $Ca^{2+}$ parameters were measured, including **H** Frequency of $Ca^{2+}$ transients, **I** $Ca^{2+}$ transient recovery time, **J** normalized $Ca^{2+}$ transient amplitude (mean ± SEM, *$p = 0.0129$, WT $n = 10$, plna R14del $n = 12$), **K** normalized diastolic $Ca^{2+}$ levels, and **L** Tau of $Ca^{2+}$ decay in wild-type (WT) and plna R14del embryonic zebrafish (mean ± SEM, *$p = 0.0498$, WT $n = 10$, plna R14del $n = 12$). All measurements were performed in three experimental replicates. Wild type is highlighted in black and plna R14del in red. Statistical test: unpaired Student's t-test. nM nanomolar, ms milliseconds, CaM calmodulin, UAS upstream activation sequence, $Ca^{2+}$ calcium. Source data are provided as a Source Data file.

Together these results demonstrate that istaroxime rescues the APD alternans in plna R14del mutant cardiomyocytes and indicates that these APD alternans are caused by defects in intracellular $Ca^{2+}$ handling.

## Discussion

In this study, we utilized the zebrafish to investigate the in vivo consequence of a PLN R14del mutation. Our data indicate that plna R14del results in alterations of cardiac intracellular $Ca^{2+}$ dynamics already at the embryonic stages, and that these changes relate to a reduction in SR $Ca^{2+}$ re-uptake. At this early age, the observed defects in $Ca^{2+}$ handling do not lead to measurable differences in cardiac output. In the adult heart plna R14del cardiomyocytes start to show alterations in APD, especially under high pacing frequencies, presumably as a consequence of reduced SERCA2a activity. In line with this, the hearts of adult plna R14del zebrafish displayed clear variations in cardiac output. In elderly (>1 year old) plna R14del mutant fish we observed cardiac remodeling, which was characterized by the accumulation of immune cells, fibroblast and fat, features reminiscent of those seen in PLN R14del patients. Importantly, our results demonstrate that drug treatment with istaroxime improves cardiac $Ca^{2+}$ dynamics, enhances cardiac relaxation and reverses APD alternans in plna R14del cardiomyocytes.

We observed a heterogeneous effect of plna R14del in 2-year-old zebrafish, as some fish appeared completely healthy while others displayed end-stage heart disease characterized by the accumulation of fibroblasts, immune cells and fat in the sub-epicardial region. A similar heterogenous effect of the PLN R14del mutation is seen in patients, with fibrofatty infiltrates that are usually located in the posterolateral region of the left ventricular myocardium, right underneath the epicardium. Cardiac fibrofatty replacement in PLN R14del patients has a likely (sub) epicardial origin, which implicates that the underlying mechanisms of disease might be comparable between zebrafish and man[37]. The epicardium is a mesothelial cell layer that acts as an important source of trophic signals to maintain continued growth and differentiation of the developing heart. The importance of the epicardium as a driver of morphological changes in the heart is increasingly recognized[38]. In the adult zebrafish the epicardium responds to injury by inducing embryonic epicardial gene expression (e.g. tbx18) and proliferation[27,39]. After cardiac injury, the epicardial cells undergo a process called epithelial-to-mesenchymal transformation (EMT). They migrate into the cardiac tissue, where they can give rise to cardiac smooth muscle cells and fibroblasts[40]. Epicardial cells can also give rise to fat cells via mesenchymal transformation and peroxisome proliferator-

activated receptor γ (PPARγ) activation[41]. As the remodeled tissue within plna R14del hearts showed a strong expression of tbx18, it suggests that it may have an epicardial origin. This should be studied further by lineage tracing experiments.

Interestingly, plna R14del zebrafish presented with a remarkable higher variation in ventricular outflow peak velocity, visible as alternating strong and weak beads in the echocardiography data. Pulsus alternans result primarily from an alternating contractile state of the ventricle[42]. Importantly, a recent clinical study revealed that pre-symptomatic PLN R14del carriers display mechanical alterations in the left ventricle and that this change in mechanical properties precedes structural remodeling[43]. This is consistent with our observation that mechanical variations are observed in non-remodeled plna R14del zebrafish hearts. In accordance with our echocardiography measurements, we noticed a clear alternans between consecutive APs in plna R14del cardiomyocytes during patch clamp experiments at the high pacing frequency of 4 Hz. Alternans may be a membrane voltage or AP-driven phenomenon or due to $Ca^{2+}_I$[31,44]. In the present study, we found no differences in average APD in a wide range of frequencies between WT and plna R14del myocytes, which is supported by the virtually overlapping of major ion currents underlying in the cardiac AP. This strongly argues against a membrane voltage or AP-driven mechanisms of our observed APD alternans[31]. A plausible mechanism underlying the alternans is a functional change in SR $Ca^{2+}$ pumps, resulting in alternating strong and weak beats[44,45]. Also, concomitant slow transportation of $Ca^{2+}$ from the uptake compartment to the release compartment in the SR has been suggested as a cause of alternans[44]. Corroborating such a model, we observed a slower $Ca^{2+}$ reuptake in plna R14del cardiomyocytes and by application of istaroxime we were able to restore reduced $Ca^{2+}$ amplitudes and decay phase, and prevent electrical alternans. Istaroxime is a drug that acts by releasing PLN from SERCA and thereby stimulates SR $Ca^{2+}$ loading and release[34]. We did not find $Ca^{2+}$ transient alternans, possibly because we were not able to reach the required 4 Hz stimulation with the use of field stimulation during the measurement of cytoplasmic $[Ca^{2+}]$ after Indo-1 loading in adult myocytes.

In line with our presumptions, the in vivo $Ca^{2+}$ transient amplitude decreased in embryonic plna-R14del mutant fish and the $Ca^{2+}$ reuptake duration was clearly prolonged, potentially caused by a hampered SR $Ca^{2+}$ sequestration[6]. While the diastolic $Ca^{2+}$ concentration was unaffected in embryonic hearts of plna R14del mutants, it was increased in adult plna R14del cardiomyocytes. Importantly, in embryonic zebrafish hearts ncx1 is strongly expressed and loss of Ncx1 activity results in $Ca^{2+}$ overload[46]. Differences in embryonic and adult ncx1 expression

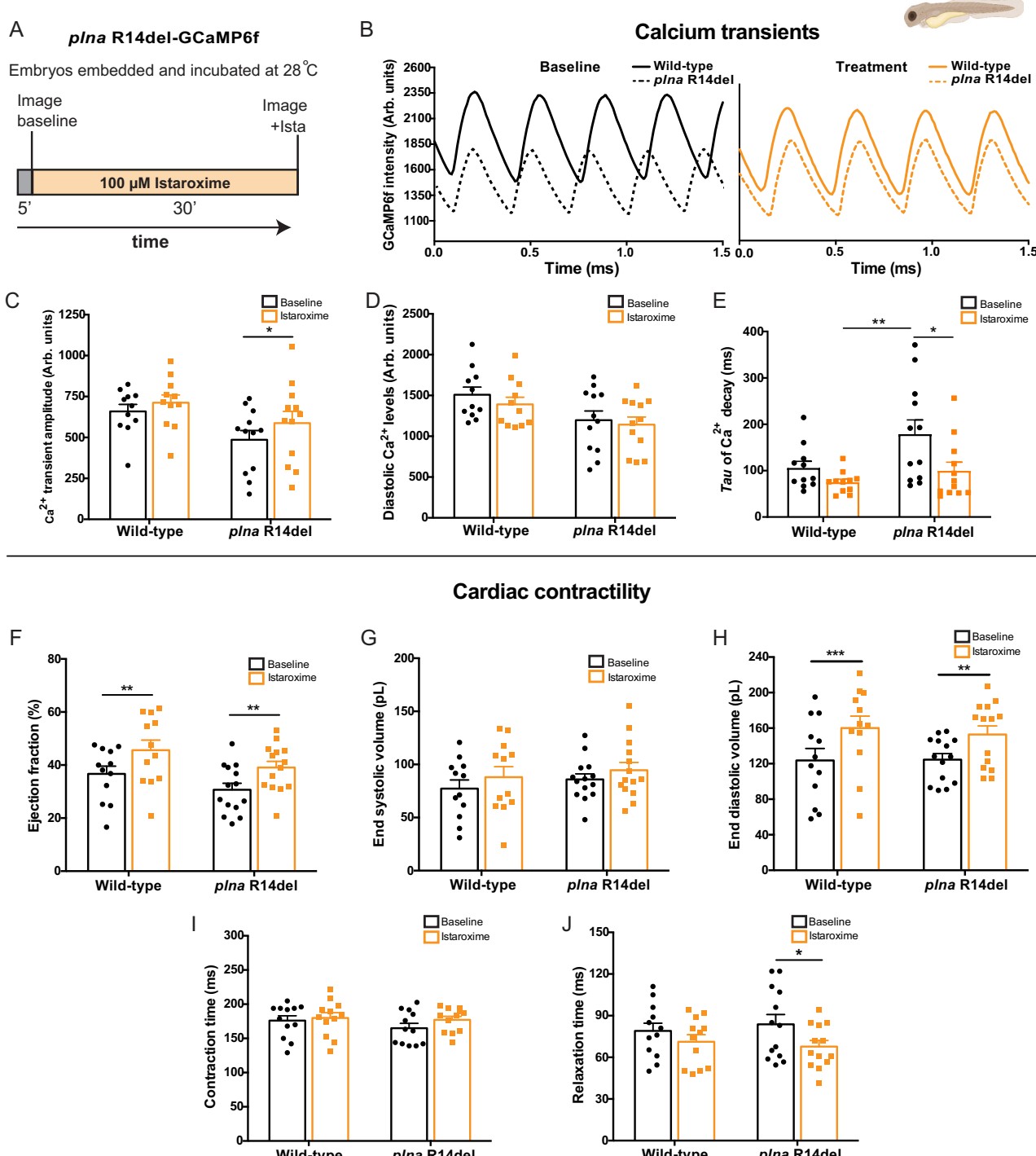

may therefore explain observed differences in diastolic Ca$^{2+}$ levels in embryonic and adult *plna* R14del hearts. In line with literature, we demonstrated that istaroxime improves intracellular Ca$^{2+}$ transient amplitude and has a positive inotropic effect on zebrafish heart function. More efficient reuptake of Ca$^{2+}$ into the SR, by mitigated PLN-SERCA inhibition, will improve SR Ca$^{2+}$ load, Ca$^{2+}$ release, and as a consequence elevates the Ca$^{2+}$ transient amplitude. The enhanced Ca$^{2+}$ transient amplitudes did not directly translate into improved contractility of the zebrafish cardiomyocyte in our study. This may be explained by high

myofilament affinity for Ca2+, which may lead to complete myofilament saturation in both the absence and presence of istaroxime. Zebrafish have the ability to adapt their myofilament Ca$^{2+}$ sensitivity extensively in order to maintain cardiac function under a large range of temperatures (6–38 °C), demanding a high Ca$^{2+}$ (ref. [47]). In this study, we show that istaroxime predominantly improves the relaxation, relaxation time, and filling of the zebrafish heart, rather than affecting the contractile state of the heart. The positive effect of istaroxime on cycle length relaxation time was primarily present in *plna* R14del mutants.

**Fig. 6 Istaroxime improves cardiac calcium dynamics and cardiac relaxation in _plna_ R14del embryonic zebrafish. A** Overview of the experimental setup. Embryos were embedded in agarose and incubated for 5 min at 28 °C. Baseline measurements were performed, and subsequently, embryos were incubated with 100 µM istaroxime for 30 min, and imaged again. **B** Representative calcium ($Ca^{2+}$) transients from one wild type (WT) and one _plna_ R14del embryonic zebrafish, before and after incubation with 100 µM istaroxime. **C** $Ca^{2+}$ transient amplitude (mean ± SEM, *$p = 0.0324$, WT baseline $n = 11$, WT istaroxime $n = 11$, _plna_ R14del baseline $n = 12$, _plna_ R14del istaroxime $n = 13$, two-way ANOVA). **D** Diastolic $Ca^{2+}$ levels and **E** Tau of $Ca^{2+}$ decay in WT and _plna_ R14del zebrafish, at baseline and after incubation with 100 µM istaroxime (mean ± SEM, *$p = 0.0439$, **$p = 0.0049$, WT baseline $n = 11$, WT istaroxime $n = 11$, _plna_ R14del baseline $n = 12$, _plna_ R14del istaroxime $n = 13$, two-way ANOVA). **F–J** Bar graphs of contractility parameters at baseline and after incubation with 100 µM istaroxime, including end-systolic volume **F** (mean ± SEM, **$p = 0.005988$ for WT and **$p = 0.005588$ _plna_ R14del, WT baseline $n = 12$, WT istaroxime $n = 12$, _plna_ R14del baseline $n = 14$, _plna_ R14del istaroxime $n = 15$, two-way ANOVA), end-diastolic volume **G**, ejection fraction **H** (mean ± SEM, ***$p = 0.0002$, **$p = 0.0015$, WT baseline $n = 11$, WT istaroxime $n = 11$, _plna_ R14del baseline $n = 12$, _plna_ R14del istaroxime $n = 13$, two-way ANOVA), contractile cycle length contraction time **I** and contractile cycle length relaxation time **J** (mean ± SEM, *$p = 0.0006$, WT baseline $n = 11$, WT istaroxime $n = 11$, _plna_ R14del baseline $n = 12$, _plna_ R14del istaroxime $n = 13$, two-way ANOVA). All measurements were performed in three experimental replicates. Statistical test two-way ANOVA. Baseline condition is highlighted in black and treated in orange. Arb. units arbitrary units, ms milliseconds, pL picoliter. Source data are provided as a Source Data file.

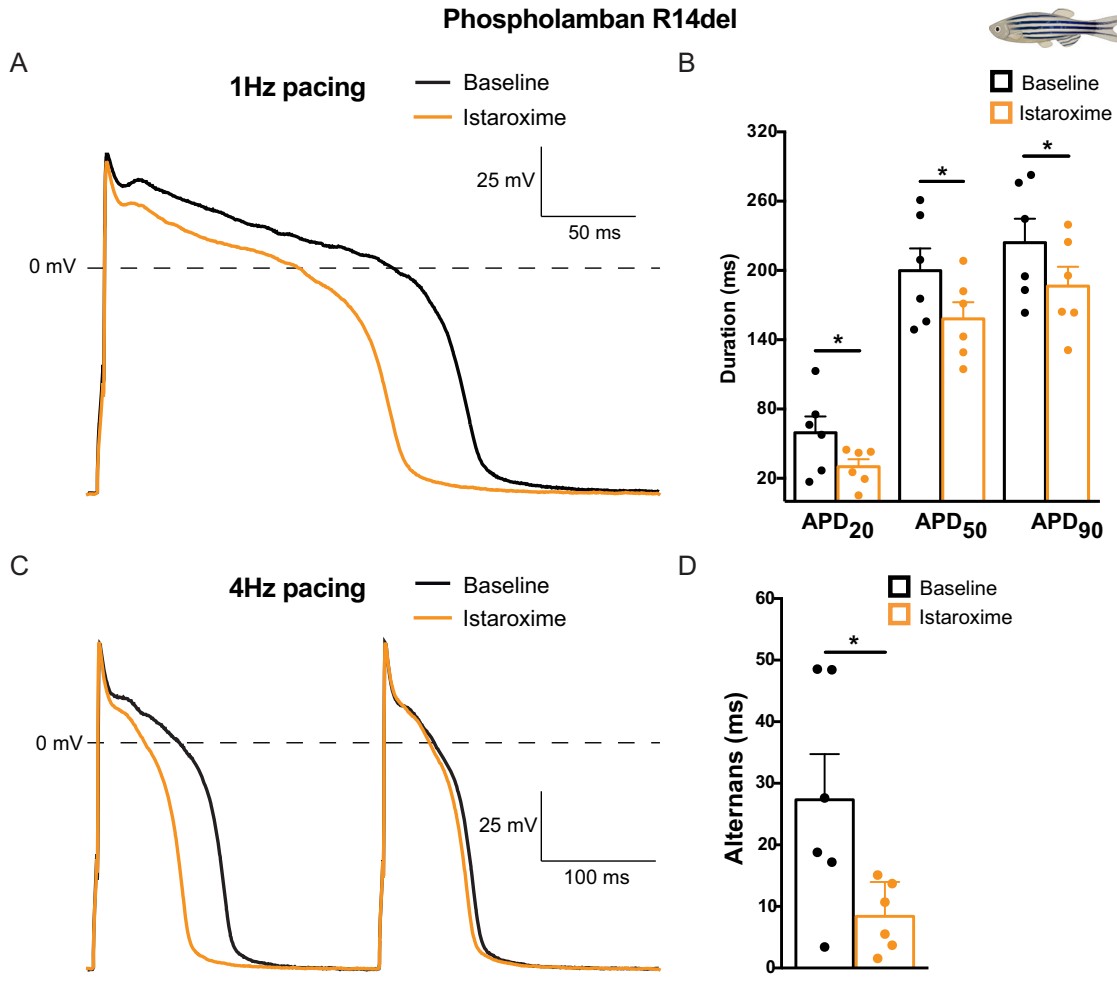

**Fig. 7 Istaroxime reverses action potential alternans in _plna_ R14del isolated adult cardiomyocytes. A** Typical action potentials (APs) of _plna_ R14del isolated adult cardiomyocytes at baseline (black line) and after incubation with 5 µM istaroxime (orange line), paced at 1 Hertz (Hz). **B** Average AP parameters of _plna_ R14del isolated adult cardiomyocytes at baseline (black bars) and after incubation with 5 µM istaroxime (orange bars), at 1 Hz pacing (mean ± SEM, $APD_{20}$: *$p = 0.030$, $APD_{50}$: *$p = 0.022$, $APD_{90}$: *$p = 0.020$, baseline $n = 6$, istaroxime $n = 6$, paired Student's $t$-test). **C** Typical action potentials (APs) of _plna_ R14del isolated adult cardiomyocytes at baseline (black line) and after incubation with 5 µM istaroxime (orange line), paced at 4 Hz. The _plna_ R14del cardiomyocyte showed an AP alternans at baseline, where a long AP is followed by a short AP. **D** Average $APD_{90}$ difference between two consecutive APs (alternans) at baseline and after incubation with 5 µM istaroxime, during 4 Hz pacing (mean ± SEM, *$p = 0.039$, baseline $n = 6$, istaroxime $n = 6$, paired Student's $t$-test). All measurements were performed in three experimental replicates. Baseline condition is highlighted in black and treated in orange. Hz Hertz, mV millivolt, $APD_{20}$ AP duration at 20% of repolarization, $APD_{50}$ AP duration at 50% of repolarization, $APD_{90}$ AP duration at 90% of repolarization. Source data are provided as a Source Data file.

These data in combination with the clearly reduced $Ca^{2+}$ transient *tau* of decay confirm that istaroxime can improve SR $Ca^{2+}$ re-uptake[34]. Ouabain is a widely used inotropic agent in heart failure therapy. Unlike istaroxime, ouabain did not affect the $Ca^{2+}$ transient amplitudes. This suggests a predominant lusitropic effect of istaroxime via enhanced SR $Ca^{2+}$ sequestration. Clinical trials with 6–24 h istaroxime infusion in heart failure patients, however, did not demonstrate improved left ventricular diastolic volumes[48,49], which is expected from improved SR $Ca^{2+}$ reuptake. This may be explained by the Na/K-ATPase inhibiting activity of istaroxime, which leads to increased diastolic $Ca^{2+}$.

This study has also demonstrated that istaroxime shortens AP repolarization in the zebrafish heart and isolated zebrafish *pln* R14del cardiomyocytes, a phenomenon previously reported in vitro and in vivo in the chronic atrioventricular block dog model[50]. Under physiological conditions, the removal of $Ca^{2+}$ from the cytosol is achieved via SR $Ca^{2+}$ reuptake and via the forward mode of the NCX. Upon istaroxime exposure, the enhanced SR sequestration of $Ca^{2+}$ through SERCA2A during diastole likely reduces the amount of $Ca^{2+}$ extruded by the NCX and reduces its depolarizing pump current. A reduced amplitude of NCX current, in combination with enhanced $Ca^{2+}$-induced $I_{Ca,L}$ inactivation, therefore results in shortening of repolarization. Impaired NKA $Na^+$ extrusion by istaroxime elevates intracellular $Na^+$ levels and favors the reverse mode of NCX, further shortening AP repolarization. More important, in this study we show that APD alternans caused by the *plna* R14del mutation can be reversed. This highlights $Ca^{2+}$ handling defects as underlying cause for the APD alternans in isolated ventricular *plna* R14del cardiomyocytes. APD alternans are likely not related to a prolonged repolarization, as $APD_{90}$ did not differ between wild type and *pln* R14del cardiomyocytes. We appreciate that the $Ca^{2+}$ handling defects in *plna* R14del embryonic hearts and adult cardiomyocytes cannot yet be linked directly towards the structural remodeling at later stages. In *PLN* R14del patients, the disease symptoms predominantly manifest at later ages (50–60 years of age), implying that APD alternans and impaired $Ca^{2+}$ transient dynamics are an early phenotypical characteristic of the disease, which over time predisposes to a pathogenic $Ca^{2+}$ handling related cardiomyopathy.

Some limitations of this study should be considered. First, the *pln* gene has been duplicated in the zebrafish due to the teleost-specific genome[22,51]. Duplicated genes can act redundant or may diverge in their function, which may complicate interpolations to mammalian models. As both *plna* and *plnb* are highly expressed in the ventricle and our studies show their genetic interaction, we favor the hypothesis that *plna* and *plnb* have redundant roles in the zebrafish ventricle. A second limitation is that istaroxime used in this study has a dual function as it stimulates SERCA2 activity and inhibits $Na^+/K^+$ ATPase (NKA) activity[35,36]. The *plna* R14del zebrafish model may help to screen for more selective compounds that only improve SERCA2a activity without affecting Na/K-ATPase activity to benefit *PLN* R14del carriers.

In conclusion, by introducing the R14del mutation in the endogenous zebrafish *plna* gene we generated a relevant zebrafish cardiomyopathy model which shows in vivo cardiac $Ca^{2+}$ dysregulation and morphological features reminiscent to those observed in cardiac specimen from patients. Our data present possible early disease mechanisms in *PLN* R14del cardiomyopathy and provide usefulness of this model to explore patient-specific drug treatment. Naturally, extrapolation of our findings to the human situation should be done with caution.

## Methods

**Zebrafish husbandry**. Fish used in this study were housed under standard conditions[52]. All experiments were conducted in accordance with the ethical guidelines and approved by the local ethics committee of the Royal Dutch Academy of Sciences (KNAW).

**Generation of mutant lines**. The R14del mutation was generated in the wild-type Tupfel Longfin (TL) strain zebrafish using CRISPR/Cas9 technology. *Plna* R14del fish were previously generated using CRISPR/Cas9 technology with homologous recombination[25]. In short, one-cell-stage zebrafish embryos were microinjected with an injection mixture consisting of (final concentrations) 150 ng/µl nuclear Cas9 (nCas9) mRNA, 20–40 ng/µl sgRNA, 10% (v/v) Phenol Red, and 25 ng/µl template oligo. Each putative founder adult fish was crossed with a wild-type adult fish (F1). Homozygous fish (F2) were generated by inbreeding heterozygous mutant carriers.

**Adult zebrafish heart isolations and preparation**. For paraffin sections, adult zebrafish hearts were dissected and fixed in 4% paraformaldehyde (dissolved in phosphate buffer containing 4% sucrose) at 4 °C overnight, washed twice in PBS, dehydrated in EtOH, and embedded in paraffin. Serial sections were made at 10 µm using a microtome (Leica, RM2035). For cryosections, zebrafish hearts were extracted and fixed in 4% paraformaldehyde (in phosphate buffer) for 4 h at room temperature (RT). Three washes of 30 min were performed using 4% sucrose (in phosphate buffer) followed by overnight incubation at 4 °C in 30% sucrose (in phosphate buffer). Hearts were embedded in tissue freezing medium (Leica, Lot# 03811456), frozen on dry ice and kept at −80 °C. Cryo-sectioning using Cryostar NK70 (Thermo Scientific) was performed to obtain 10 µm thin sections. Images of extracted whole hearts were acquired using a Leica M165 FC stereo microscope.

**Adult zebrafish heart staining**. ISH was performed on paraffin sections according to standard protcol[53], with the exception that the hybridization buffer did not contain heparin and yeast total RNA. Briefly, slides were dehydrated, tissue was digested with proteinase K and then (pre-) hybridized with the probe of interest overnight. After a series of washes with SSC, SCC with formamide, and 0.1% PBS-Tween 20, slides were incubated in α-DIG-AP (Roche, 11093274910) overnight for antibody detection. Staining was developed using NBT-BCIP (Roche, 11681451001). Primer sequences for the *grn1, postnb* probes are shown in Table S1. The *myl7* and *tbx18* ISH probes were previously generated[54,55]. Hematoxylin and eosin (H&E) staining was performed on cryosections and paraffin sections according to the standard laboratory protocol, with the exception that cryosections were fixed in 4% paraformaldehyde for 1 h at RT beforehand. Picro-sirius Red staining on cryosections and paraffin sections was performed in accordance with the standard laboratory protocol. Briefly, dewaxed and hydrated the paraffin sections, stained nuclei in hematoxylin for 8 min, washed slides in running water for 10 min, and stained with picro-sirius red (Sigma Aldrich, 365548-5G Direct Red 80) for 1 h. Then, washed 2× in acidified water (5 ml acetic acid (Merck, M8792) in 1 L of water), removed all water by vigorous shaking of slides, dehydrated 3× in 100% EtOH, cleared in xylene before mounting. Oil Red O (ORO) staining was performed on cryosections according to the standard protocol[56,57]. Briefly, cryosections slides were first air dried and rinsed in 70% ethanol. Slides were stained with Oil Red O solution (Sigma Aldrich, 00625) for 2–3 h, rinsed in 70% ethanol and then distilled water. Counter staining was performed with hematoxylin for 5 min, and washed the well in distilled water. Imaging of stained sections was performed using a Leica DM4000 B LED upright automated microscope. Terminal dUTP nick-end labeling (TUNEL) apoptosis staining was performed on cryosections and detected using In Situ Cell Death Detection Kit, fluorescein from Roche (Mannheim, Germany, Lot#29086800) according to the manufacturer's instructions. Nuclei were stained with DAPI (4′,6-diamidino-2-phnylindole) from Molecular Probes. Confocal images were acquired using a Leica Sp8 confocal microscope and processed using Imaris image analysis software (version 9.3.1).

**Echocardiography**. Adult zebrafish were anesthetised with 16 mg/ml of MS-222 in aquarium water. When the zebrafish were unresponsive to a slight tail pinch with a forceps, it was determined that the appropriate anesthetised condition for echocardiography was reached. Anesthetized zebrafish were placed in a custom-made mold and placed ventral side up in a large petridish filled with 4% MS-222 in aquarium water. Breathing was monitored by visual tracking of opercular movement to ensure fish health throughout the protocol. Transthoracic echocardiography was performed using a Vevo2100 Imaging System (VisualSonics Inc., Toronto, Canada) with a 50 MHz ultrasound transducer fixed above the ventral side of the zebrafish and parallel to the longitudinal axis plane. Recordings were performed within 3 min after induction of anesthesia to preserve optimal cardiac performance. Quantitative measurements were assessed offline using the Vevo2100 analytical software. Color Doppler images were used for measuring heart rate, ventricular outflow diameter (manually), ventricular outflow tract velocity time integral (VOT VTI) and ventricular outflow peak velocity (VOT PV), as validated by Wang et al.[58] and Visual Sonics Inc. Every parameter was determined as the average measurement of at least five cardiac cycles. Ventricular outflow surface was calculated by the equation $\pi \times (0.5 \times OT \text{ diameter})^2$; VOT diameter was measured in three consecutive beats per fish. Stroke volume was calculated by multiplying the VOT VTI with the ventricular outflow surface. Variation in outflow peak velocity was determined by the discrepancy in VOT PV between two consecutive heart

beats and averaged over at least six beats per fish. VOT PV variation was corrected for the average total OT PV per fish. Hearts of these fish were then individually isolated and processed as frozen samples for generation of cryosections.

**Cellular electrophysiology.** Single ventricular cardiomyocytes were isolated by an enzymatic dissociation procedure as used previously for zebrafish sinoatrial node and atrial cells[59]. Here, ventricles from three to four adult fishes (6–7 months) were pooled and stored at RT in a modified Tyrode's solution containing (in mmol/l): NaCl 140, KCl 5.4, CaCl$_2$ 1.8, MgCl$_2$ 1.0, glucose 5.5, HEPES 5.0; pH 7.4 (set with NaOH). Subsequently, the ventricles were cut in small pieces, which were transferred to Tyrode's solution with 10 μmol/l CaCl$_2$ (30 °C). The solution was refreshed one time before the addition of Liberase TM research grade (final concentration 0.038 mg/ml (Roche Diagnostics, GmbH, Mannheim, Germany)) and Elastase from porcine pancreas (final concentration 0.01 mg/ml (Bio-Connect B.V., Huissen, Netherlands)) for 12–15 min. During the incubation period, the tissue was triturated through a pipette (tip diameter: 2.0 mm). The dissociation was stopped by transferring the ventricular pieces into a modified Kraft–Brühe solution (30 °C) containing (in mmol/l): KCl 85, K$_2$HPO$_4$ 30, MgSO$_4$ 5.0, glucose 5.5, pyruvic acid 5.0, creatine 5.0, taurine 30, β-hydroxybutyric acid 5.0, succinic acid 5.0, BSA 1%, Na$_2$ATP 2.0; pH 6.9 (set with KOH). The tissue pieces were triturated (pipette tip diameter: 0.8 mm) in Kraft–Brühe solution (30 °C) for 4 min to obtain single cells. Finally, the cells were stored for at least 45 min in modified Kraft–Brühe solution before they were transferred into a recording chamber on the stage of an inverted microscope (Nikon Diaphot), and superfused with Tyrode's solution (28 °C). Quiescent single cells with smooth surfaces were selected for electrophysiological measurements. APs and net membrane currents were recorded using the amphotericin-B perforated patch-clamp technique and an Axopatch 200B amplifier (Molecular Devices, Sunnyvale, CA, USA). Voltage control and data acquisition were realized with custom-made software Scope (version 04.04.27; kindly provided by J. Zegers) and analysis was performed with the custom-made software, MacDaq (version 10.7.1; kindly provided by A. van Ginneken). Pipettes (resistance 3–4 MΩ) were pulled from borosilicate glass capillaries (Harvard Apparatus, UK) using a custom-made microelectrode puller, and filled with solution containing (in mmol/l): K-gluconate 125, KCl 20, NaCl 10, amphotericin-B 0.44, HEPES 10; pH 7.2 (set with KOH). Potentials were corrected for the calculated liquid junction potential[60]. Signals were low-pass-filtered with a cut-off of 5 kHz and digitized at 40 and 5 kHz for APs and membrane currents, respectively. Cell membrane capacitance ($C_m$) was estimated by dividing the time constant of the decay of the capacitive transient in response to 5 mV hyperpolarizing voltage clamp steps from –40 mV by the series resistance. APs were elicited at 0.2–4 Hz by 3 ms, ~1.2× threshold current pulses through the patch pipette. Susceptibility to delayed afterdepolarization (DAD) generation was tested using fast burst pacing (20 APs at 3 Hz) which was followed by an 8-sec pause. After the 8 s pause, a single AP was evoked to test the susceptibility of early afterdepolarizations generation. APs were characterized by RMP, maximum AP amplitude (APA$_{max}$), AP duration at 20, 50 and 90% of repolarization (APD$_{20}$, APD$_{50}$, APD$_{90}$ respectively), maximal velocity (d$V$/d$t$) of the AP upstroke (Phase-0) and phase-3 repolarization (Phase-3), and plateau amplitude (APA$_{plat}$; measured 50 ms after the AP upstroke). Averages were taken from 10 consecutive APs. DADs were defined as spontaneous depolarization of >1 mV. The AP measurements were alternated by a general voltage clamp protocol to elucidate the ionic mechanism underlying the AP changes. For K$^+$ current measurements, 500 ms depolarizing and hyperpolarizing voltage clamp steps were applied from a holding potential of –50 mV with a cycle length of 2 s (Fig. S3A). To ensure that the other cardiomyocytes in the recording chamber remained undistorted for biophysical analysis, our voltage clamp measurements were performed without specific channel blockers or modified solutions. Inward rectifier K$^+$ current ($I_{K1}$) and rapid delayed rectifier K$^+$ current $I_{Kr}$ were defined as the quasi-steady-state current at the end of the voltage-clamp steps at potentials negative or positive to –30 mV, respectively. The L-type Ca$^{2+}$ current ($I_{Ca,L}$) was measured with a two-pulse voltage clamp protocol (Fig. S3B) from a holding potential of –60 mV. The first pulse (P1) served to activate $I_{Ca,L}$; the second pulse (P2) was used to analyze the inactivation properties of $I_{Ca,L}$. $I_{Ca,L}$ was defined as the difference between peak current and steady-state current. Current densities were obtained by normalizing to $C_m$. After baseline recordings, isolated ventricular cardiomyocytes were treated for 5 min with 5 μM istaroxime (MedChemExpress, Lot#11394). The effect on APD was measured at 1 and 4 Hz. [Ca$^{2+}$]$_i$ were measured at 25 °C in HEPES solution ((mmol/l): [Na$^+$] 156, [K$^+$] 4.7, [Ca$^{2+}$] 1.3, [Mg$^{2+}$] 2.0, [Cl$^-$] 150.6, [HCO$_3^-$] 4.3, [HPO$^{2-}$] 1.4, [HEPES] 17, [Glucose] 11, and 1% fatty acid free albumin, pH 7.3) using the fluorescent probe Indo-1 as described previously[61]. In brief, isolated myocytes were exposed to 5 μmol/l of the acetoxymethyl esters of indo-1 during 30 min at 25 °C. Myocytes were attached to a poly-D-lysine (0.1 g/l)-treated coverslip placed on a temperature-controlled microscope stage of an inverted fluorescence microscope (Nikon Diaphot) with quartz optics. A temperature-controlled perfusion chamber (height 0.4 mm, diameter 10 mm, volume 30 μl, temperature 25 °C), with two needles at opposite sides for perfusion purposes, was tightly positioned over the coverslip. The contents of the chamber could be replaced within 100 ms. Bipolar square pulses for field stimulation (40 V/cm) were applied through two thin parallel platinum electrodes at a distance of 8 mm. One quiescent single myocyte was selected (myocytes with more than one spontaneous oscillation per 10 s were

excluded) and the measuring area was adjusted to the cell surface with a rectangular diaphragm. The wavelength of excitation of Indo-1 was 340 nm, applied with a stabilized xenon-arc lamp (100 W). Fluorescence was measured in dual emission mode at 410 and 516 nm. Emitted light passed a barrier filter of 400 nm, a dichroic mirror (450 nm) and respective narrow band interference filters in front of two photomultipliers (Hamamatsu R-2949). Signals were digitized at 1 kHz and corrected for background signals recorded from Indo-1-free myocytes. Ten subsequent Ca$^{2+}$ transients were averaged from which apparent [Ca$^{2+}$]$_i$ was calculated according to the ratio equation[62].

**High-speed brightfield imaging.** Embryos were placed in 1-phenyl-2-thiourea (PTU) 20–24 h post fertilization (hpf) to prevent pigmentation. Three days post-fertilization (dpf) embryos were embedded in 0.3% agarose prepared in E3 medium containing 16 mg/ml MS-222. Recordings were performed at 150 frames per seconds (fps) using a high-speed inverted light microscope at 28 °C. Basal parameters were recorded first. Subsequently 3 ml of 100 μM istaroxime (MedCham Exptress, Lot#11394) or ouabain (Sigma-Aldrich, Lot#BCBZ9329) in E3-MS-222 solution was added, incubated for 30 min, and parameters were measured for a second time. Heart rate measurements and contractility parameters were analyzed using ImageJ (U.S. National Institutes of Health, Bethesda, Maryland, USA). Contraction time, relaxation time, and contraction cycle time were analyzed by scrolling through the recorded movies frame by frame, and by identifying (1) the moment the ventricular wall moved inward and the ventricle started expelling blood (start of contraction); (2) the moment the ventricular wall moved back outward (start of relaxation); and (3) the moment that the ventricular wall reached its most dilated position (maximum relaxation). This process was repeated three times for each heart and values were averaged. The hemodynamic parameters such as surface area and volumes were analyzed using ImageJ by drawing an ellipse on top of the ventricle at end-diastole and end-systole. Per heart six ellipses were analyzed: three at diastole and three at systole. Values were averaged. ImageJ provided the values for the minor and major axis of each ellipse. Surface area was calculated using the following formula: (0.5 × major axis) × (0.5 × minor axis) × π. End-diastolic and end-systolic volume (EDV/ESV) were calculated by: (1/6) × (π) × (major axis) × (minor axis$^2$). Stroke volume (SV) by EDV−ESV. Ejection fraction (EF) by SV/EDV. Cardiac output (CO) by SV × heart rate.

**High-speed fluorescence imaging.** *PLN* R14del fish were crossed to tg(myl7:-Gal4FF; UAS:GCaMP6f) fish[33] to obtain *PLN* R14del fish with a genetically encoded cardiac Ca$^{2+}$ sensor. Wild-type fish expressing GCaMP6f or a genetically encoded voltage sensor (VSFP Butterfly CY) were used as controls[33]. A morpholino (MO) oligomer targeted against *tnnt2a* (5′-CATGTTTGCTCTGATCTGA-CACGCA-3′) from Genetools was injected at the one-cell stage to uncouple contraction from excitation in embryos, thereby preventing motion artifacts in our recordings of intracellular cardiac Ca$^{2+}$ handling[63]. Embryos were placed in PTU after 20–24 h to keep them transparent. In all, 3 dpf embryos were embedded in 0.3% agarose prepared in E3 medium containing 16 mg/ml MS-222 and placed in a heated (28 °C) recording chamber. Recordings were performed with Micro-manager 1.4.17 (Image software) using a custom-built upright widefield microscope (Cairn Research) equipped with a ×20 1.0 NA objective (Olympus XLUMPLFLN20X W). Blue LED excitation light (470 nm) was filtered using a 470/40 nm filter (Chroma ET470/×40) and reflected towards the objective using a 515 nm dichroic mirror (Chroma, T515lp). Emitted fluorescence was filtered by a 514 long-pass filter (Semrock, LP02-514RU) and images were projected on a high-speed camera (Andor Zyla 4.2 plus sCMOS). Recordings were performed at 100 fps for 1000 frames. Basal parameters (heart rate, Ca$^{2+}$ transient amplitude, diastolic Ca$^{2+}$ level, upstroke time, recovery time) were recorded first. Subsequently, drug stocks were diluted in 28 °C E3-MS-222 medium (istaroxime: MedChemExpress, Lot#11394, thapsigargin: Sigma-Aldrich, T9033, ouabain: Sigma-aldrich, Lot#BCBZ9329) and the medium was mixed vigorously to assure a homogeneous concentration of the drug. Embryos were incubated for 30 min in the E3-MS222-istaroxime mixture and parameters were measured again. Recordings were analyzed using ImageJ and Matlab (Version R2015a, Mathworks).

**Quantitative PCR.** RNA was isolated from hearts of adult TL wild-type fish. Adult hearts were separated into ventricular ($n = 3$) and atrial ($n = 3$) samples. ef1α was used as a reference gene for the qPCR. Samples were loaded on a 96-well PCR plate and run in a CFX real-time PCR System (Bio rad) using KiCStart SYBR Green qPCR ReadyMix with ROX as recommended by the manufacturer (Sigma-Aldrich: KCQS02). Three technical replicates were performed for each sample together with no-template control (NTC) for each gene. The qPCR was performed with a 2-min hold at 50 °C and a 10 min hot start at 95 °C, followed by the amplification step for 43 cycles of 15 s denaturing at 95 °C and 1 min annealing/extension at 58 °C, and the melting curve was obtained by increments of 1 °C/ 10 s from 63 to 95 °C. Primers for qPCR are described in detail in Table S1. Analysis was performed on the Cq values as previously shown by calculating fold change using the formula FC = 2$^{-(ΔΔCt)}$, which is based on calculating the averaged changed between the *pln* values in relation to *ef1α*, the reference gene[64].

**Electrocardiography (ECGs)**. Adult zebrafish were anesthetized for 5 min in 16 mg/ml MS-222 in aquarium water and placed ventrally on a submerged sponge. ECG recordings were performed using a local field potential recording chamber as described previously[65]. For electrode placement, a grounding wire (1) was inserted in the submerged sponge, the reference electrode (2) placed on the abdominal tissue and recording electrode, and silver wire placed in a glass capillary filled with 1 mM NaCl (3) was placed on the heart region. Data were amplified using a DAGAN EX-1 amplifier, digitized with a National Instruments USB-6210 and recorded using LabscribeNI. Signals were amplified 100× and a 3-100 Hz bandpass filter was applied. Recordings lasted 3 min in total and heart rate was calculated using the LabscribeNI PRS detector function. Single Labscribe files were exported and Matlab was used to calculate average ECG complexes to generate a single wild type and mutant trace on which interval and amplitude analysis could be performed.

**Statistical analysis**. Statistical analysis and drawing of graphs and plots were carried out in GraphPad Prism (version 6 for Mac OS X and version 7 for Windows, GraphPad Software) and SigmaStat 3.5 software. Normality and equal variance assumptions were tested with the Kolmogorov–Smirnov and the Levene median test, respectively. Differences between two groups were analyzed using the paired Student's $t$-test, comparisons between experimental groups were analyzed by one-way ANOVA for non-parametric variables with Tukey's post-test for intergroup comparisons. Comparisons between experimental groups in combination with an intervention were analyzed by two-way ANOVA for non-parametric variables with Tukey's post-test for intergroup comparisons. All data are presented as mean ± SEM, and $p < 0.05$ was considered significant. $*p \leq 0.05$, $**p \leq 0.01$, $***p \leq 0.001$, $****p \leq 0.0001$, n.s. $p > 0.05$. All comparisons with no statistical indications on the figures are non-significant. $N$ denotes the number of fish used per dataset.

**Reporting summary**. Further information on research design is available in the Nature Research Reporting Summary linked to this article.

## Data availability
The authors declare that all data supporting the findings of this study are available within the paper and its supplementary information. Any remaining raw data will be available from the corresponding author upon reasonable request. Source data are provided with this paper.

## Code availability
The analysis scripts used in this study (Peaks, https://osf.io/86ufe/) have been made available with instructions and sample data on Gitlab (https://gitlab.com/OVeth/sarah/-/tree/master).

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

## Acknowledgements
We would like to thank the Hubrecht Institute animal care takers for fish care, Anko de Graaff at the imaging facility, Jeroen Korving and Harry Begthel at the histology department. We acknowledge the support from The Netherlands Cardio Vascular Research Initiative (CVON): the Dutch Heart Foundation, Dutch Federation of University Medical Centres, the Netherlands Organization for Health Research and Development and the Royal Netherlands Academy of Sciences (CVON-PREDICT 2012-10, 2018-30) and the PLN Foundation. This work was further supported by the ZonMW grant 40-00812-98-12086, the ERA-NET Cofund action No. 643578 under the European Union's Horizon 2020 research and innovation program and national funding organizations Canadian Institutes for Health Research (CIHR), the Netherlands Organization for Health Research and Development (ZonMw), Belgium (Flanders) Research Foundation Flanders (FWO), French National Research Agency (ANR) and E-Rare-CoHeart project.

## Author contributions
S.M.K.: study design, experiments in adult animals, data analysis and interpretation, manuscript preparation, reviewing and editing, expansion and care of the animal colony. C.J.M.v.O.: study design, experiments in embryonic and adult animals, data analysis and interpretation, manuscript preparation, reviewing and editing, care of the animal colony. C.D.K.: generation of the animal model, study design, experiments in embryonic and adult animals, data analysis and interpretation, manuscript preparation, reviewing and editing, expansion and care of the animal colony. A.O.V.: patch clamp experiments, data interpretation and analysis, manuscript reviewing and editing. B.d.J.: adult cardiomyocyte isolations. B.J.D.B.: calcium measurements on adult cardiomyocytes, data interpretation and analysis, manuscript reviewing and editing. Y.L.O.: cryo-sectioning and care of the animal colony. E.v.A.: cryo-sectioning and staining of adult heart section. S.C.: generation and expansion of the animal colony C.P.P.: In vivo calcium imaging in embryonic animals. W.J.W.: ECG experiment, data interpretation and analysis, manuscript reviewing and editing. M.A.V.: funding, manuscript reviewing and editing. T.P.d.B.: conceptual contribution and study design, data analysis and interpretation, manuscript reviewing and editing. T.A.B.v.V.: study conceptualization, main funding and design, data analysis and interpretation, study coordination, manuscript reviewing, and editing. J.B.: study conceptualization, generation of the animal model, main funding and design, data analysis and interpretation, study coordination, manuscript reviewing and editing.

## Competing interests
The authors declare no competing interests.
