## [Peer Review File · Nature Communications]

Reviewers' comments:

Reviewer #1 (Remarks to the Author):

In this study, a *plna* R14del mutation previously associated with dilated or arrhythmogenic cardiomyopathy is evaluated in a zebrafish model. The authors find some evidence of action potential and Ca²⁺ handling abnormalities along with structural abnormalities compared to wild type. This topic would be interesting to readers of the Journal, and the manuscript is well written and easy to understand. However, there are some concerns that need to be addressed.

The data do not strongly support the conclusions drawn. For example, it is concluded that *plna* R14del results in alterations of cardiac intracellular Ca²⁺ dynamics already at the embryonic stages, and that these changes relate to a reduction in SR Ca²⁺ re-uptake (lines 484-485). If this were true, then the decay phase of the Ca²⁺ transient should be affected, but it was not. Additionally, reduced SR Ca²⁺ re-uptake is associated with Ca²⁺ transient alternans, however this was not observed. Saying that istaroxime rescues or restores impaired Ca²⁺ to wild type levels is not supported by the data (lines 439-440), because no comparison was made to wild type values (Figure 6C). Given that istaroxime decreased APD and APD alternans, does not mean this was by its effect on Ca²⁺ handling (lines 478-479). APD shortening by itself can reduce APD alternans independent of any changes in Ca²⁺ handling.

Some of the results are inconsistent. For example, there is APD alternans but no Ca²⁺ transient alternans in the *plna* R14del group. Usually these coexist. Also, istaroxime decreased relaxation time but had no effect on Ca²⁺ transient recovery. Usually these track each other. Compared to numerous previous reports, if Ca²⁺ re-uptake is decreased (slowed), then Ca²⁺ transient alternans should be observed, but they were not. Finally, it's hard to believe that ouabain has no effect on wt Ca²⁺ transient amplitude and contraction, because it has been previously shown in calibrated Ca²⁺ transient amplitudes. (PMID 23203972).

One important claim is that functional remodeling appears before structural remodeling. However, different methods are used at different times. To strengthen this claim, similar experiments should be performed at similar stages. For example, Ca²⁺ handling should be assessed at older stages (e.g. *ex vivo* hearts or isolated myocytes). This may explain why some of the results are inconsistent.

While istaroxime may be a reasonable treatment option, its use as a tool to dissect mechanisms is very limited because it is non-specific.

It's possible that some of the results may be an artifact of the methods. For example, genetic Ca²⁺ indicators have a very slow response time. Could this have affected the results? Also, for the istaroxime experiments Ca²⁺ imaging was performed before and then after 30 min of drug. Time controls should be performed to confirm that no change in calcium fluorescence occurs after 30 min. Finally, Ca²⁺ signals were not calibrated. This is a problem because absolute fluorescent levels can vary depending on experimental conditions (e.g. excitation light). Figure 5E shows normalized calcium transient amplitudes; however, it's not clear what the signals were normalized to – baseline fluorescence? Give the importance of the Ca²⁺ transient amplitude results, Ca²⁺ transients must be calibrated.

The structural results shown in figure 2 are excellent and very convincing. The results in figure 1 are also good; however, the differences between groups are not obvious. It may be helpful to quantify these results to make a stronger argument. Furthermore, if the Oil Red O staining is striking, then I would include that in the main manuscript. This issue is especially important to strengthen claims regarding the sub-epicardial involvement.

The pulses alternans described in figure 3 is not obvious. First, there is no obvious alternans in Panel F (Figure 3). Second, alternans is quantified by variations, which could be random. It would be best to compare the average strong beat and average weak beat for each group. Finally, the H&E results shown in figure 3G that show no difference in hearts with strong pulses alternans, may not be the best approach because the differences shown in Figure 1 (a positive control) are not very strong either. Maybe a different stain (e.g. those used in figure 2) would have been better for showing these differences.

Need to indicate that AP alternans was measured during steady state pacing and not when pacing is first turned on.

Reviewer #2 (Remarks to the Author):

This well-written manuscript describes the in-vivo embryonic and adult phenotypes, incl from isolated adult cardiomyocytes, of a zebrafish PLN R14del cardiomyopathy model. This important study adds to the growing body of work highlighting the utility of adult zebrafish cardiomyopathy

and arrhythmia models, which display age-related functional and structural changes that closely resemble human pathologies, to discover disease mechanisms and potential treatments that promise translation into medicine. The development of functional models such as this, which can be used to discover and test novel approaches to treatment (as demonstrated here with Istaroxime), is essential to fulfil the promise of personalised medicine.

Major comments:

1. Generation of the disease model:

Page 11, lines 316-321: The existence of two or more zebrafish orthologues of a given human gene is a common problem faced by zebrafish cardiac researchers. Could the authors please comment further on their rationale for modelling the human heterozygous PLN R14del mutation using zebrafish with homozygous *plna* R14del/R14del and wild-type *plnb* +/+, rather than double heterozygous *plna* R14del/+ and *plnb* R14del/+?

2. Structural findings in adult *plna* R14del/R14del zebrafish:

Figure 2: could the authors please comment on the apparent expansion of epicardial cells, as well as infiltration with immune cells, into the bulbus arteriosus, which appears morphologically altered, see Figure 1 (with possible functional effects on outflow, see 3. below).

In some images in Figures 1, 2 and S2 the atrium seems altered/remodelled/enlarged, which is surprising given the lack of *plna* expression in the atrium?

Te Rijdt et al, 2019, describe that human PLN R14del cardiomyocytes are characterised by distinct desmosomal protein distributions, including of SAP97 and GSK3b. Have the authors considered conducting similar immunofluorescence experiments on isolated adult zebrafish cardiomyocytes to further corroborate similarities between human and zebrafish models?

3. Functional abnormalities in adult *plna* R14del/R14del zebrafish:

Page 12, line 368, Page 6, lines 175-178, and Figure 3: could the authors please explain their rationale for using “Color Doppler images to examine contractile parameters”, rather than typical B-mode images, analysed using the Visualsonics LV trace tool or strain analysis to obtain ejection fraction, stroke volume and cardiac output? Ideally this information should be added to the current data. Color and Pulse-wave Doppler-based determination of outflow is dependent on factors such as imaging plane and morphology of the bulbus arteriosus (which appears altered in mutant fish in Figures 1 and 2). The pulse-wave (not color) Doppler traces shown Figure 3F show a typical elliptic shape and regular spacing of the outflow envelope in WT fish, while the shape of the outflow envelope is more “shark fin-like” in mutants, which may indicate suboptimal positioning of the sampling window or imaging plane. Was this a typical finding? Did the bulbus arteriosus seem normal on B-mode echocardiography?

The presence of variation in peak outflow velocity could be illustrated more clearly in Figure 3F, maybe using horizontal markers (the red and yellow bar is difficult to see).

Adult ECGs would have been a useful addition to complete the picture.

5. Phenotypical heterogeneity in adult *plna* R14del/R14del zebrafish and isolated cardiomyocytes.

Some morphological abnormalities and also functional graphs in Figure 4 (dV/dt and alternans) suggest the presence of normal and abnormal populations of fish/cells. Did the authors find that these correlated, ie. those fish with abnormal histology also showed functional cardiac alternans on echo and electric alternans in individual cardiomyocytes?

Could the authors speculate on the reasons for variable phenotype penetrance? In patients with R14del mutation, additional genetic variants or environmental factors are suggested to influence phenotypical heterogeneity, but presumably these fish are all siblings and their environment is relatively tightly controlled?

4. Embryonic phenotyping:

Findings of impaired Ca²⁺-dynamics are interesting in the context of the developing cardiac conduction system, which at 3dpf is still immature. (eg. Chi et al. Genetic and physiologic dissection of the vertebrate cardiac conduction system. PLoS Biol. 2008 May 13;6(5):e109. doi: 10.1371/journal.pbio.0060109; Yu et al. Evolving cardiac conduction phenotypes in developing zebrafish larvae: implications to drug sensitivity. Zebrafish 2010 Dec;7(4):325-31. doi: 10.1089/zeb.2010.0658.) Have the authors attempted further studies at later developmental time points, and considered embryonic/larval ECG?

Minor comments:

Page 6, lines 163 and 166: could the authors please check final MS-222 concentration is correct?

Page 8, line 231: could the authors please specify the number of isolated cardiomyocytes studied.

Page 15, line 442: reference 46 is incorrect at this position, as this paper does not describe istaroxime use in acute heart failure

Page 15, line 455: please change potentiates to potentiate

Figure S2: missing panels for TUNEL staining under images E and F

Figure S8, panel B, bottom right panel on APD: please change $n=0.06$ to $p=0.06$ (ns) or similar

Reviewer #3 (Remarks to the Author):

The paper from Kamel et al. describes a zebrafish model for the p.Arg14del mutation which is associated with dilated/arrhythmogenic cardiomyopathy in humans. They go on to show that cardiac remodeling and fibrosis in elderly fish is preceded by contractile dysfunction and diminished Ca²⁺ transients - a potential early disease mechanism, and importantly that istaroxime treatment can rescue the Ca²⁺ dysregulation.

The R14del is a Dutch founder mutation which accounts for up to 15% of all Dutch patients with DCM/ACM, and the mutation has also been seen in other population groups, highlighting the potential impact of a therapeutic option where currently no specific drugs exist, and the utility of an in vivo model which recapitulates the human phenotype.

Major points

1. The authors generate a knock in mutation for the R14del in Plna and then go on to suggest, given the presence of the Plnb homologue, zebrafish with a homozygous mutation in Plna and wildtype plnb are representative of the human heterozygotes. However it is not clear that the two homologues have completely overlapping functions, it is likely there is some redundancy but as the authors already show there are differences in expression of the homologs between ventricle and atrium - in particular the very high expression of plnb in the atrium (fold change around 200).
2. The analysis is based on the R14del plna mutant, but can the authors be sure there weren't any crispr mediated off target effects that could also partly explain the phenotype. Ideally the same R14del would need to be made in the plnb on a plna wild type background and at the very least show similar changes in Ca²⁺ dynamics.

Minor points

- whilst investigating whether cardiac morphological changes occur earlier the authors say they identified one case with a similar altered tissue organisation (Figure S2). It is hard to recognise the heart - did the authors stain for any cardiac/ventricular markers?

- formatting of references should be checked e.g. ref 14, 23, 24, 25

Reviewer #4 (Remarks to the Author):

The submitted manuscript by Kamel et al. is an elegant study targeting a rare form of genetic cardiomyopathy (heterozygous phospholamban (PLN) p.Arg14del (R14del) mutation) affecting patients in the Netherlands and some other countries.

The mutation was introduced in their zebrafish model using CRISPR/Cas9 technology.

The authors detect a phenotype in the mutated fish hearts that nicely mimicks the patterns observed in patients with the mutation.

Moreover, they report found that istaroxime treatment

ameliorates the in vivo Ca²⁺ dysregulation, rescues the cellular APD alternans, while it improves cardiac relaxation.

It is concluded that these effects are due to an interaction of istaroxime with PLB as reported previously in several papers. It should be acknowledged that the the displacement of PLB from SERCA2 has been proposed as a mechanism of action but never been directly proven. The data in favour of this mechanism almost entirely derives from papers written or sponsored by the producer of that compound.

A direct measurement of phospholamban or changes of its function are also lacking in the submitted manuscript. The improvement of cardiac function is shown by effects on Ca transients, contractility and APDs. One would assume decreasing diastolic Ca upon SERCA stimulation, which was not detected. This might be due to the properties of the dye used but should be discussed. Secondly, the larger end diastolic volume seen after istaroxime seems to fit to SERCA-activation mechanisms but is also seen unchanged in the mutated fish.

Overall, the effects of istaroxime might also be explainable by sole Na/K-ATPase inhibition.

A potential way to challenge or prove the model and hypothesis might be to perform Ca transient measurements at different and higher stimulation rates (as with the APDs). Sole Na/K-ATPase inhibitors would likely worsen diastolic function whereas additional SERCA stimulation might prevent this. A potential comparator might be cardiac glycosides.

Human data has shown no different effect of these two compounds in vitro (Wallner et al. 2017) and the clinical trial performed with istaroxime did neither show decrease in diastolic tension nor end-

diastolic elastance compared to placebo but a decrease in end diastolic volume not observed in placebo (HORIZON-HF Shah 2009). The clinical program with istaroxime was stopped with two more clinical trials being withdrawn and a third one with unreported status (clinicaltrials.gov).

This human data should also be discussed in the manuscript.

Reviewer #1 (Remarks to the Author):

In this study, a *plna* R14del mutation previously associated with dilated or arrhythmogenic cardiomyopathy is evaluated in a zebrafish model. The authors find some evidence of action potential and Ca²⁺ handling abnormalities along with structural abnormalities compared to wild type. This topic would be interesting to readers of the Journal, and the manuscript is well written and easy to understand. However, there are some concerns that need to be addressed.

We thank the reviewer for his/her time and effort to review our work, their positive comments and constructive remarks. Below, the points raised by the reviewer are indicated in normal font, followed by our point-by-point response in blue.

The data do not strongly support the conclusions drawn. For example, it is concluded that *plna* R14del results in alterations of cardiac intracellular Ca²⁺ dynamics already at the embryonic stages, and that these changes relate to a reduction in SR Ca²⁺ re-uptake (lines 484-485). If this were true, then the decay phase of the Ca²⁺ transient should be affected, but it was not.

We thanks the reviewer for raising this point. We now have performed additional analyses and experiments that further support this conclusion and added these data to the revised manuscript. First, to study the decay phase of the Ca²⁺ transient in more detail, we quantified the *tau* of decay from Ca²⁺ transients of *plna* R14del mutants and wild type siblings using an exponential fit, at baseline and after treatment with Istaroxime. We found that the decay phase is significantly slower in *plna* R14del mutants indicated by the increased tau (WT: 105.74 ± 14.35 ms and *plna*-R14del: 178.6 ± 30.78 ms, p≤0.05) and that in both *plna* R14del mutants and wild-type siblings (more prominent in mutants) istaroxime clearly results in a faster decay phase as indicated by a reduced *tau* of decay. These data are now presented in Figures 5L and 6E.

Second, to better understand Ca²⁺ dynamics in the embryonic heart and the consequence of SERCA2a inhibition, we have treated *tg(myf7:Gal4FF UAS:GCaMP6f)* embryos with the SERCA inhibitor thapsigargin. These results are now presented in Supplementary Figure S7 and show that that SERCA inhibition results in a significant lowering of the Ca²⁺ transient amplitudes and elongation of the recovery time.

Third, we now quantified absolute values for intracellular Ca²⁺ concentration ([Ca²⁺]_i) in ventricular myocytes isolated from adult fish. We used calcium indicator Indo-1 and measured fluorescence in a bichromatic manner. The latter allows for calculating absolute diastolic and systolic [Ca²⁺]_i. We found that diastolic [Ca²⁺]_i was elevated in *plna* R14del cardiomyocytes. Moreover, transient duration was longer and the decay phase was slower as indicated by an increased *tau* (WT: 368.05 ± 142.50 ms and *plna*-R14del: 764.88 ± 86.61 ms, p≤0.05), which is consistent with impaired SR Ca²⁺ uptake due to the *plna* R14del mutation. These new results are presented in Fig. 5B-F.

Together, new data on the *tau* of decay in embryonic fish, the thapsigargin experiments and the quantification of absolute [Ca²⁺]_i in adult cardiomyocytes are consistent with a conclusion that *plna* R14del impaires SR Ca²⁺ uptake. These description of these results are included in the manuscript on page 15.

Additionally, reduced SR Ca²⁺ re-uptake is associated with Ca²⁺ transient alternans, however this was not observed. Saying that istaroxime rescues or restores impaired Ca²⁺ to wild type levels is not supported by the data (lines 439-440), because no comparison was made to wild type values (Figure 6C).

As suggested by the reviewer, we now made comparisons to wild-type values. We performed statistical analysis on the data shown in Figure 6C using two-way Anova. This revealed that there is no significant difference between Ca^{2+} transient amplitudes of *plna* R14del embryos with istaroxime and wild type embryos. The only significant difference in Ca^{2+} transient amplitudes was found between untreated *plna* R14del embryos and *plna* R14del embryos with istaroxime. Hence, we conclude that istaroxime restores Ca^{2+} transient amplitudes to wild-type levels. In addition the new analysis described above demonstrate that istaroxime also rescues the prolonged Ca decay time in the PLN R14del embryos (shown in Figure 6E).

Given that istaroxime decreased APD and APD alternans, does not mean this was by its effect on Ca^{2+} handling (lines 478-479). APD shortening by itself can reduce APD alternans independent of any changes in Ca^{2+} handling.

The reviewer raises a relevant issue. She/he is right that the istaroxime-induced APD shortening by itself may reduce APD alternans, but in our *plna* R14del cardiomyocytes, we think that the APD alternans is related to altered Ca^{2+} homeostasis rather than APD duration for the following reasons: In basal conditions, the average APD is similar in both WT and *plna* R14del myocytes in a wide range of frequencies. This is supported by the virtually overlapping of major ion currents involved in the action potential. The absence of average APD differences between wild-type and *plna* R14del cardiomyocytes strongly argues against a membrane voltage or action potential (AP)-driven mechanisms (Kulkarni et al., 2019; PMID: 31617437) of the APD alternans in the present study. This is further supported by the fact that APD alternans under basal conditions only occurred in *plna* R14del cardiomyocytes at the fastest tested pacing rate, which typically have the shortest APD (Figure 4C of the revised manuscript). If the APD was the main cause of the alternans, it also would occur at slower pacing rates with longer APs. In basal conditions, however, we found Ca^{2+} abnormalities, which are suggested as a cause of alternans (Kihara & Morgan, 1991, PMID: 1750531; Kulkarni et al., 2019, PMID: 31617437). We therefore think that the restoration of Ca^{2+} homeostasis by istaroxime is the main cause for the APD alternans reduction.

Some of the results are inconsistent. For example, there is APD alternans but no Ca^{2+} transient alternans in the *plna* R14del group. Usually these coexist.

These differences are likely due to differences in methodology. APD alternans were measured in single cells, while the Ca^{2+} transients in the original manuscript were analysed in vivo. These differences prevent direct comparison of in vitro and in vivo results. For example, the in vivo analysis was performed on whole hearts, in which cells are coupled via gap junctions and thereby stabilise each other. In addition, the presence of an active autonomic nervous system in the 3 dpf embryo may modulate Ca^{2+} dynamics and action potential duration. We now performed additional Ca^{2+} measurements in isolated cardiomyocytes (see above). Unfortunately, the use of field stimulation during these Ca^{2+} measurements prevent from driving the cardiomyocytes faster than 3 Hz. Accordingly, we did not observe any alternans.

Also, istaroxime decreased relaxation time but had no effect on Ca^{2+} transient recovery. Usually these track each other. Compared to numerous previous reports, if Ca^{2+} re-uptake is decreased (slowed), then Ca^{2+} transient alternans should be observed, but they were not. Finally, it's hard to believe that ouabain has no effect on wt Ca^{2+} transient amplitude and contraction, because it has been previously shown in calibrated Ca^{2+} transient amplitudes. (PMID 23203972).

As mentioned above, we have now characterized the Ca^{2+} transients in more detail and found that the decay phase is significant slower in *plna* R14del mutants in both embryo's and isolated adult

cardiomyocyte (shown in Figure 5) and that istaroxime restores the decay phase in PLN R14del embryos (see Figure 6). We observed alternans in APs at 4 Hz, but not in Ca^{2+} transients in embryos and isolated adult cardiomyocytes due to the technical issue mentioned in the previous comment. We do not have a forward explanation for the lack of effects of ouabain on Ca^{2+} transients in the present study, but we cannot exclude that it is related to our model, i.e., *in vivo* embryonic zebrafish, while other findings were obtained from *ex vivo* isolated adult mouse ventricular myocytes.

One important claim is that functional remodeling appears before structural remodeling. However, different methods are used at different times. To strengthen this claim, similar experiments should be performed at similar stages. For example, Ca^{2+} handling should be assessed at older stages (e.g. *ex vivo* hearts or isolated myocytes). This may explain why some of the results are inconsistent.

To address this, we have now included additional data:

In Figure 3G-L we included *mRNA in situ* hybridization for markers of immune cells and fibroblasts on adult *plna* R14del hearts that showed function alternans with echocardiography. These new results confirm that functional changes are detected prior to any structural changes in these hearts.

As suggested by the reviewer, we now measured $[Ca^{2+}]_i$ in ventricular cardiomyocytes isolated from adult fish using the calcium indicator Indo-1, and these results are displayed in Figure 5 and described above. These results are directly comparable to the action potential measurements, which were performed from adult cardiomyocytes as well.

Importantly, during the revision phase, a clinical study was published in which pre-symptomatic PLN R14del carriers were subjected to echocardiography with a 3-year follow-up. In this study it was concluded that mechanical alterations in the left ventricle preceded structural remodeling, which is consistent with our observations in *plna* R14del zebrafish. We have added this to the discussion on page 18 (lines 557-560):

“Importantly, a recent clinical study revealed that pre-symptomatic *PLN* R14del carriers display mechanical alterations in the left ventricle and that this change in mechanical properties precedes structural remodeling⁵⁶. This is consistent with our observation that mechanical alterations are observed in non-remodeled *plna* R14del zebrafish hearts”.

While istaroxime may be a reasonable treatment option, its use a tool to dissect mechanisms is very limited because it is non-specific.

We are aware of the dual function of istaroxime, as it not only an enhancer of SERCA2a, but additionally is known to inhibit the Na^+/K^+ ATPase (NKA) activity. In order to further specify the observed effects obtained with istaroxime, we therefore performed additional experiments with ouabain, a specific NKA inhibitor (see Figure S11). On page 16 (500-512) we describe the effects of ouabain in comparison to those obtained with istaroxime:

“Since istaroxime is known to block NKA and potentiate SERCA2a activity, we tested whether istaroxime improves the Ca^{2+} transient amplitude via SERCA2a activation or NKA inhibition. Therefore, we compared the effect of istaroxime and ouabain, a specific NKA blocker, on ventricular Ca^{2+} transient dynamics in wild-type embryos. In contrast to istaroxime, ouabain treatment had no effect on Ca^{2+} transient amplitude (Figure S11A&B). To test whether a NKA block could be responsible for the lusitropic effects of istaroxime, potential changes in contractility were examined in ouabain-treated embryos. Ouabain did not significantly enhance stroke volume, ejection fraction or cardiac output (Figure S11C-F). Next, we validated the effectiveness of ouabain in our zebrafish model using embryos expressing a genetically encoded Voltage Sensitive Fluorescent Protein (VSFP Butterfly-CY) (Figure S12A). Fluorescent high-speed

recordings of VSFP embryos demonstrated a clear APD shortening in the presence of ouabain compared to baseline values (Figure S12B).

*Together, these data indicate that istaroxime can improve SR Ca²⁺ reuptake in *plna* R14del mutant cardiomyocytes, an effect that is predominantly asserted via interference of the PLN-SERCA interaction”*

It's possible that some of the results may be an artifact of the methods. For example, genetic Ca²⁺ indicators have a very slow response time. Could this have affected the results?

We agree with the reviewer that genetic Ca²⁺ indicators have a slower response time compared to Ca²⁺ sensitive dyes. In addition, genetic indicators are sensitive to alterations in expression levels. Therefore, we now include Ca²⁺ measurements using the Ca²⁺ sensitive dye Indo-1 (see above) and observed that Ca²⁺ reuptake is slower in *plna* R14del myocytes, using both methods. The difference in diastolic Ca²⁺ measured by either the genetic sensor and the Ca²⁺-sensitive dye are not well explained by a difference in methodology but more likely reflect differences in Ca²⁺ handling during embryonic and adult stages. We have added a few lines to the discussion to highlight this (page 19, lines 572-575), which read:

*“Importantly, in embryonic zebrafish hearts *ncx1* is strongly expressed and loss of Ncx1 activity results in Ca²⁺ overload⁶⁰. Differences in embryonic and adult *ncx1* expression may therefore explain observed differences in diastolic Ca²⁺ levels in embryonic and adult *plna* R14del hearts.”*

Also, for the istaroxime experiments Ca²⁺ imaging was performed before and then after 30 min of drug. Time controls should be performed to confirm that no change in calcium fluorescence occurs after 30 min

We thank the reviewer for this addressing this. Placebo experiments to examine changes in fluorescent intensity over time (30 min) have been performed and are shown in Fig. R1.1 below. From these results we conclude that the factor time has no effect on Ca²⁺ transient parameters. Some of this data has already been published when we described the generation of the genetic sensor tools (van Opbergen CJM, Koopman CD et al., Theranostics, 2018, pmid 30279735).

Figure R1.1. Examination of action potential and Ca²⁺ transients in 3 dpf zebrafish embryos carrying a transgenic calcium sensor. (A) Action potential frequency and parameter at baseline and after 30 minutes. (B) Frequency of Ca²⁺ transients, calcium transient amplitudes and diastolic Ca²⁺ levels. Statistics: mean ± SEM, n.s. not significant, WT n=15, *plna* R14del n=16, unpaired Students t-test). min⁻¹: per minute, ms: milliseconds, , APD₉₀: AP duration at 90% of repolarization.

Finally, Ca²⁺ signals were not calibrated. This is a problem because absolute fluorescent levels can vary depending on experimental conditions (e.g. excitation light). Figure 5E shows normalized calcium transient amplitudes; however, it's not clear what the signals were normalized to – baseline fluorescence? Give the importance of the Ca²⁺ transient amplitude results, Ca²⁺ transients must be calibrated.

We agree that absolute fluorescent levels can vary depending on experimental conditions. We did not calibrate the GCamp6 fluorescent signal with another fluorescent signal as this was not available. For that reason, we performed all experiments with the same settings for the microscope and lighting source. In addition, measurements performed on the same day were on mixed clutches of control and *plna* R14del larvae so that all would be imaged under very similar conditions. After imaging the individual larvae, these were genotyped (for wild-type *plna* or *plna* R14del) and the Ca²⁺ transients were analysed blindly to prevent any bias. After this, the blinding was removed and the

analysed Ca^{2+} data was separated into wild-type and *plna* R14del groups. We reasoned that this procedure should minimize the influence of technical variations on the outcome of the results. As we work with absolute numbers (Arbitrary Units), we normalised the values to calculate the percentage of change. Values are normalised to the mean of the entire group (WT/ KO/w drug/wo drug).

As described above we now also included Ca^{2+} measurements on isolated myocytes using the Ca^{2+} sensitive Indo-1 dye and a ratiometric calculation method to derive absolute intracellular $[\text{Ca}^{2+}]$. These results are now included in Figure 5B-F.

The structural results shown in figure 2 are excellent and very convincing. The results in figure 1 are also good; however, the differences between groups are not obvious. It may be helpful to quantify these results to make a stronger argument. Furthermore, if the Oil Red O staining is striking, then I would include that in the main manuscript. This issue is especially important to strengthen claims regarding the sub-epicardial involvement.

To better show the differences between the remodelled and non-remodelled *plna* R14del^{-/-} hearts we have now changed the magnifications of the images in Figure 1. In the previous version of this figure we had reduced the size of the remodelled hearts more compared to the non-remodelled hearts (indicated by the differences in their scale bars) to fit these panels into the figure. As it might have been harder to appreciate the size differences we now show in the revised version of Figure 1 all the sections using the same magnification (except for panels A-C as the microscope and camera used where not calibrated). In addition, we have included the Oil Red O staining in the main figure (panels J-L) as suggested by the reviewer. As the cardiac remodelling phenotype in the *plna* R14del^{-/-} fish was not fully penetrant (6 out of 25) we kept a qualitative instead of a quantitative description of the phenotype. We did quantify the Sirius Red staining in the hearts of *plna* R14del/R14del;*plnb*^{-/-} fish as here the cardiac remodelling was more penetrant (22 out of 30) and included the results in Figure S4G.

The pulses alternans described in figure 3 is not obvious. First, there is no obvious alternans in Panel F (Figure 3). Second, alternans is quantified by variations, which could be random. It would be best to compare the average strong beat and average weak beat for each group. Finally, the H&E results shown in figure 3G that show no difference in hearts with strong pulses alternans, may not be the best approach because the differences shown in Figure 1 (a positive control) are not very strong either. Maybe a different stain (e.g. those used in figure 2) would have been better for showing these differences.

We have replaced Fig.3F for a more striking image and indicated the differences in the strong and weak beats for the *plna* R14del heart. We agree that there is some natural variation in beat strength (as demonstrated by the variation seen in wild-type hearts), but we observed more variation in the *plna* R14del group compared to the control group. This variation was determined by the discrepancy in VOT PV between two consecutive heart beats and averaged over at least 6 beats per fish. VOT PV variation was corrected for the average total OT PV per fish (see M&M). We have also determined the average strongest and weakest beat within the the wild type and *plna* R14del^{-/-} groups (see Figure R1.2A). This does not show significant differences between the groups, which is expected. Only when determining the differences between the strongest and weakest beat within one fish there is a clear significant difference between wild type and *plna* R14del^{-/-} fish (Figure R1.2B).

Figure R1.2. Comparison of strongest and weakest peaks in Echocardiography traces of 10 month of *plna* R14del and wild-type. (A) Peak velocities of weakest and strongest beat in *plna* R14del and wild-type (B) The difference is the strongest to weakest beat peak velocity in *plna* R14del and wild-type (mean \pm SEM, ** $p < 0.01$, wild-type $n = 21$, *plna* R14del $n = 21$, paired Students t-test). mm/s: millimetre per second.

As suggested by the reviewer we have now performed the same analysis that was used in Figure 2 to detect any structural remodelling of the 10-month-old hearts with strong pulsus alternans. These results are now presented in Figure 3G-L and are described in the results. Figure 3G-L shows expression of markers for inflammation (*grn1*), epicardial cells (*tbx18*), and fibroblast (*postnb*), which are not affected in hearts with strong pulse alternans.

Need to indicate that AP alternans was measured during steady state pacing and not when pacing is first turned on.

We thank the reviewer for pointing to this omission. In the revised version of the manuscript (p.14, line 426), we now mention that the APs are measured in steady-state conditions, including the 4 Hz pacing, which show alternans.

Reviewer #2 (Remarks to the Author):

This well-written manuscript describes the in-vivo embryonic and adult phenotypes, incl from isolated adult cardiomyocytes, of a zebrafish PLN R14del cardiomyopathy model. This important study adds to the growing body of work highlighting the utility of adult zebrafish cardiomyopathy and arrhythmia models, which display age-related functional and structural changes that closely resemble human pathologies, to discover disease mechanisms and potential treatments that promise translation into medicine. The development of functional models such as this, which can be used to discover and test novel approaches to treatment (as demonstrated here with Istaroxime), is essential to fulfil the promise of personalised medicine.

We thank the reviewer for his/her time and effort to review our work, their positive comments and constructive remarks. Below, the points raised by the reviewer are indicated in normal font, followed by our point-by-point response in blue.

Major comments:

1. Generation of the disease model:

Page 11, lines 316-321: The existence of two or more zebrafish orthologues of a given human gene is a common problem faced by zebrafish cardiac researchers. Could the authors please comment

further on their rationale for modelling the human heterozygous PLN R14del mutation using zebrafish with homozygous *plna* R14del/R14del and wild-type *plnb* +/+, rather than double heterozygous *plna* R14del/+ and *plnb* R14del/+?

We agree with the reviewer that a heterozygous *plna* R14del/+;*plnb* R14del/+ could have been an alternative strategy. We tried to introduce the R14del mutation in the *plnb* gene using the same strategy as we used to generate the *plna* R14del mutation, but were unsuccessful in doing so. We designed four Crispr sgRNA to target *plnb* at the R14 site, where two were found to be highly efficient. Next, we designed two different oligos with different homology arms lengths for each of the two CRISPR sgRNA. Each of the four combination of oligo and corresponding CRISPR sgRNAs was injected into one-cell-stage embryos. Offspring from at least 30 fish (F0) was tested for each of the different CRISPR conditions. However, no founder with the R14del mutation was identified. We don't know why this strategy was successful in introducing the R14del mutation in *plna* but not in *plnb*, but it could be due to the different genomic locations of the two *pln* genes. The *plna* and *plnb* gene encode for proteins with a high degree of homology and their levels of expression in the ventricle are very similar (see Fig. S1c). Based on this we reasoned that *plna* and *plnb* act redundantly in the ventricle and that therefore generating *plna* R14del/R14del; *plnb*+/+ fish would be a good strategy to model the heterozygous PLN R14del condition found in humans. While attempting to introduce the R14del mutation in the *plnb* gene (which failed) we did recover a *plnb* allele with a deletion of 5 base pairs resulting in a frameshift and likely loss of a functional Plnb protein. We have now collected data showing that fish homozygous for the *plna* R14del mutation and homozygous for this 5bp mutation of the *plnb* gene (*plna* R14del/R14del;*plnb*-/-) displayed more severe and earlier remodelling compared to *plna* R14del/R14del;*plnb*+/+ fish. We observe that 70% of the mutants develop the phenotype already at one year of age and that these hearts are severely remodelled. These new results confirm that *plna* and *plnb* have redundant functions in the heart. We now describe these new results at the end of page 12-13 (lines 369-377) and show the data in supplementary Figure S4.

2. Structural findings in adult *plna* R14del/R14del zebrafish:

Figure 2: could the authors please comment on the apparent expansion of epicardial cells, as well as infiltration with immune cells, into the bulbus arteriosus, which appears morphologically altered, see Figure 1 (with possible functional effects on outflow, see 3. below).

In some images in Figures 1, 2 and S2 the atrium seems altered/remodelled/enlarged, which is surprising given the lack of *plna* expression in the atrium?

Te Rijdt et al, 2019, describe that human PLN R14del cardiomyocytes are characterised by distinct desmosomal protein distributions, including of SAP97 and GSK3b. Have the authors considered conducting similar immunofluorescence experiments on isolated adult zebrafish cardiomyocytes to further corroborate similarities between human and zebrafish models?

As rightly pointed out by the reviewer, tissue remodelling around the bulbus arteriosus is apparent in most PLN R14del hearts. The epicardium is a layer of mesothelial cells that surround the heart, which has been described to be covering the atrium, ventricle and the bulbus arteriosus in zebrafish (Wang et al (2014) Nature: PMID 25938716). Wang et al demonstrated that the epicardium covering the bulbus can migrate towards the apex of the ventricle in the context of ventricular resection. This indicates that not only epicardial cells covering the ventricle but also epicardial cells covering the bulbus may be activated by injury and/or functional alterations of the ventricle. We are currently investigating the cellular origins that contribute to the epicardial thickening, which we hope to report on in a follow-up study.

The atrium is indeed remodelled in some of the pln R14del fish. We believe that the atrial remodelling is a secondary effect to the functional changes that are observed in the ventricle prior to the start of the remodelling.

The observations described by Te Rijdt et al with regard to the remodelling of desmosomal proteins including that of GSK3beta and SAP97 are not specific for the PLN R14del cardiomyopathy since remodelling of those proteins are commonly described in experimental models and patient specimen of the classical form of ACM (also known as ARVC/D) and a variety of other models mimicking cardiac diseases in which the Wnt/beta-catenin/GSK3beta pathways are affected (e.g. Asimaki PMID: 24920660, Roberts PMID: 31264976, Chelko PMID: 31533459, Tinaquero PMID: 32612162, Delmar PMID: 25446154). Also given the often poor cross-reactivity of antibodies in zebrafish we did not consider these experiments

3. Functional abnormalities in adult plna R14del/R14del zebrafish:

Page 12, line 368, Page 6, lines 175-178, and Figure 3: could the authors please explain their rationale for using “Color Doppler images to examine contractile parameters”, rather than typical B-mode images, analysed using the Visualsonics LV trace tool or strain analysis to obtain ejection fraction, stroke volume and cardiac output? Ideally this information should be added to the current data. Color and Pulse-wave Doppler-based determination of outflow is dependent on factors such as imaging plane and morphology of the bulbus arteriosus (which appears altered in mutant fish in Figures 1 and 2). The pluse-wave (not color) Doppler traces shown Figure 3F show a typical elliptic shape and regular spacing of the outflow envelope in WT fish, while the shape of the outflow envelope is more “shark fin-like” in mutants, which may indicate suboptimal positioning of the sampling window or imaging plane. Was this a typical finding? Did the bulbs arteriosus seem normal on B-mode echocardiography?

The presence of variation in peak outflow velocity could be illustrated more clearly in Figure 3F, maybe using horizontal markers (the red and yellow bar is difficult to see).

The cardiac wall of the zebrafish heart is a very thin layer of cells, which hard to visualize with conventional echocardiography imaging. In every organism (as well as zebrafish) the positioning of the heart is slightly different, and we noticed that in the far majority of zebrafish, the B-mode imaging quality was too low to detect the entire ventricular wall during the cardiac cycle of contraction. For this reason, we decided that the Ejection Fraction (based on the ventricular dimension) was impossible to examine while making use of the typical B-Mode projection. In contrast, Color and Pulse-wave Doppler imaging of cardiac outflow in these fish is very prominent and consistent between fish. For this reason, we decided that this would be the most optimal approach to examine cardiac output of the zebrafish heart.

The “shark fin-like” pulse wave seems to be more prominent in mutants, but also occurs in wildtype fish and indeed can be related to the position of the probe. In the new figure 3F, we replaced the panel now showing an example of cardiac outflow alternans in a fish with typical elliptic shape pulse waves.

We agree with the reviewer that the presence of variation could be illustrated more clearly. In a new version of Figure 3F we used horizontal instead of vertical markers.

The bulbus arteriosus is not well visible in B-mode but we performed histological analysis after the echocardiography and there was no change in the structure bulbus arteriosus of wild-type and mutant fish.

Adult ECGs would have been a useful addition to complete the picture.

We agree and we have now performed ECG recordings of 10-month old fish at baseline conditions and the results are presented in Figure S5. We were unable to detect any significant differences in the P and R amplitudes and in the P-R and P-Q intervals between wild-type and *plna* R14del fish. The T-wave could not be determined with our setup for ECGs, even with filtering the recordings for noise and other artifacts. We conclude from this, that under baseline conditions (normal heart rate), there is no effect of the *plna* R14del mutation on the electrical conduction in the heart. We have added these results to the manuscript on page 14 (lines 420-424), which read:

“To examine cardiac electrophysiology in plna R14del mutant zebrafish we first recorded electrocardiograms (ECGs) from 10-month old fish. The ECGs revealed normal PR- and PQ interval times and normal amplitudes of the P- and R-waves (Figure S5), indicating that electrical conduction from the atrium to the ventricle was unaffected in plna R14del carriers at baseline conditions..”

5. Phenotypical heterogeneity in adult *plna* R14del/R14del zebrafish and isolated cardiomyocytes. Some morphological abnormalities and also functional graphs in Figure 4 (dV/dt and alternans) suggest the presence of normal and abnormal populations of fish/cells. Did the authors find that these correlated, ie. those fish with abnormal histology also showed functional cardiac alternans on echo and electric alternans in individual cardiomyocytes?

Could the authors speculate on the reasons for variable phenotype penetrance? In patients with R14del mutation, additional genetic variants or environmental factors are suggested to influence phenotypical heterogeneity, but presumably these fish are all siblings and their environment is relatively tightly controlled?

The reviewer raises an interesting issue. We indeed observed variation in the severity of the phenotypes in individual fish. While we tried to include siblings for specific experiments, this was not always possible due to too low numbers. Furthermore, zebrafish are not inbred, so there is also genetic variation between siblings. We think that genetics can play a role in the outcome of the phenotype as we observed that fish homozygous for the *plna* R14del mutation and homozygous for a deletion of the *plnb* gene (*plna* R14del/R14del;*plnb*-/-) displayed more severe and earlier remodelling compared to *plna* R14del/R14del;*plnb*+/+ fish (see answer to question 1). The environment is indeed relatively tightly controlled.

The various experiments (morphology, Ca²⁺ measurements and electrophysiology) were all analysed in different hearts and fish, so we are unable to make correlations between the various assessed parameters. For the single-cell electrophysiological analysis we pooled hearts from different fish, so the variation observed here could be either due to cell-to-cell variation or due to variation in cells isolated from different individuals. Only for the echo measurements we also analysed the histology of the same hearts after completing the echocardiography. However, here we found that the hearts with functional alternans showed a normal histology.

4. Embryonic phenotyping:

Findings of impaired Ca²⁺-dynamics are interesting in the context of the developing cardiac conduction system, which at 3dpf is still immature. (eg. Chi et al. Genetic and physiologic dissection of the vertebrate cardiac conduction system. PLoS Biol. 2008 May 13;6(5):e109. doi: 10.1371/journal.pbio.0060109; Yu et al. Evolving cardiac conduction phenotypes in developing zebrafish larvae: implications to drug sensitivity. Zebrafish 2010 Dec;7(4):325-31. doi: 10.1089/zeb.2010.0658.) Have the authors attempted further studies at later developmental time points, and considered embryonic/larval ECG?

We agree that the heart in 3-day old larvae, when we did the Ca²⁺ recordings, is still immature. Performing the whole heart Ca²⁺ recordings at later stages is complicated as it requires crossing the

plna R14del mutation and the GCaMP6 transgene into a different genetic background that lacks pigmentation (so-called Casper fish). In this revised version of the manuscript we have extended the Ca^{2+} measurements to adult stages therefore on isolated cardiomyocytes. We measured intracellular calcium ($[Ca^{2+}]_i$) in ventricular cardiomyocytes isolated from adult fish and these results are displayed in Figure 5. These results are directly comparable to the action potential measurements, which were from adult cardiomyocytes as well. We used calcium indicator Indo-1 and measured fluorescence in a bichromatic manner. The latter allows for calculating absolute diastolic and systolic $[Ca^{2+}]_i$. We found that diastolic $[Ca^{2+}]_i$ was elevated in *plna* R14del fish. Moreover, the Ca^{2+} transient duration was significantly longer and the decay phase was significantly slower. These results are consistent with impaired SR Ca^{2+} reuptake in *plna* R14del cardiomyocytes. Unfortunately, the use of field stimulation prevents driving the cardiomyocytes faster than 3 Hz. Accordingly, we did not observe any alternans. These new results are shown in Figure 5B-F and described on page 15 (lines 447-450).

The patch clamp analysis and echocardiography were performed on adult fish, with mature cardiomyocytes and a mature conduction system. As described above we performed ECGs also by using adult fish. ECGs were not performed on embryos/larvae as these signals were too weak to analyse.

Minor comments:

Page 6, lines 163 and 166: could the authors please check final MS-222 concentration is correct?
We thank the reviewer for pointing to this error, which is now corrected to: 16mg/ml

Page 8, line 231: could the authors please specify the number of isolated cardiomyocytes studied.
The number of cells analysed is provided in the legend of Figure 7. Line 235 (methods) was adjusted to avoid confusion.

Page 15, line 442: reference 46 is incorrect at this position, as this paper does not describe istaroxime use in acute heart failure
The reviewer is right and the reference has been removed

Page 15, line 455: please change potentiates to potentiate
This has been changed

Figure S2: missing panels for TUNEL staining under images E and F
The panels are included in the revised figure. We did not observe any positive TUNEL signal in the wild type hearts, therefore the panels under E are black. In the *plna* R14del hearts, we did observe TUNEL positive cells but only in the outer remodeled layer of the heart (indicated with arrows in panels under F) and not in the myocardium.

Figure S8, panel B, bottom right panel on APD: please change $n=0.06$ to $p=0.06$ (ns) or similar
Changed to $p=0.06$ (ns). Note that old figure S8 is now S12

Reviewer #3 (Remarks to the Author):

The paper from Kamel et al. describes a zebrafish model for the p.Arg14del mutation which is associated with dilated/arrhythmogenic cardiomyopathy in humans. They go on to show that cardiac remodeling and fibrosis in elderly fish is preceded by contractile dysfunction and diminished Ca^{2+} transients - a potential early disease mechanism, and importantly that istaroxime treatment can rescue the Ca^{2+} dysregulation.

The R14del is a Dutch founder mutation which accounts for up to 15% of all Dutch patients with

DCM/ACM, and the mutation has also been seen in other population groups, highlighting the potential impact of a therapeutic option where currently no specific drugs exist, and the utility of an in vivo model which recapitulates the human phenotype.

We thank the reviewer for his/her time and effort to review our work, their positive comments and constructive remarks. Below, the points raised by the reviewer are indicated in normal font, followed by our point-by-point response in blue.

Major points

1. The authors generate a knock in mutation for the R14del in *Plna* and then go on to suggest, given the presence of the *Plnb* homologue, zebrafish with a homozygous mutation in *Plna* and wildtype *plnb* are representative of the human heterozygotes. However it is not clear that the two homologues have completely overlapping functions, it is likely there is some redundancy but as the authors already show there are differences in expression of the homologs between ventricle and atrium - in particular the very high expression of *plnb* in the atrium (fold change around 200).

The *plna* and *plnb* gene encode for proteins with a high degree of homology and their levels of expression in the ventricle are very similar (see Fig. S1c). Based on this we reasoned that *plna* and *plnb* act redundantly in the ventricle and that therefore generating *plna* R14del/R14del; *plnb*+/+ fish would be a good strategy to model the heterozygous PLN R14del condition found in humans. We also tried to introduce the R14del mutation in the *plnb* gene using the same strategy as we used to generate the *plna* R14del mutation, but were unsuccessful in doing so. We designed four Crispr sgRNA to target *plnb* at the R14 site, where two were found to be highly efficient. Next, we designed two different oligos with different homology arms lengths for each of the two CRISPR sgRNA. Each of the four combination of oligo and corresponding CRISPR sgRNAs was injected into one-cell-stage embryos. Offspring from at least 30 fish (F0) was tested for each of the different CRISPR conditions. However, no founder with the R14del mutation was identified. We don't know why this strategy was successful in introducing the R14del mutation in *plna* but not in *plnb*, but it could be due to the different genomic locations of the two *pln* genes.

While attempting to introduce the R14del mutation in the *plnb* gene (which failed) we did however recover a *plnb* allele with a deletion of 5 base pairs resulting in a frameshift and likely loss of a functional *Plnb* protein. We observed that fish homozygous for the *plna* R14del mutation and homozygous for the 5 bp deletion in the *plnb* gene (*plna* R14del/R14del;*plnb*-/-) displayed more severe and earlier remodelling compared to *plna* R14del/R14del;*plnb*+/+ fish. We observed that 70% of the *plna* R14del/R14del;*plnb*-/- fish develop the heart remodelling phenotype already at the age of 1 year. These new results are consistent with *plna* and *plnb* playing redundant roles in the heart. We now describe these new results on page 12 (lines 369-377) and show the data in supplementary Figure S4.

2. The analysis is based on the R14del *plna* mutant, but can the authors be sure there weren't any crispr mediated off target effects that could also partly explain the phenotype. Ideally the same R14del would need to be made in the *plnb* on a *plna* wild type background and at the very least show similar changes in Ca²⁺ dynamics.

The reviewer asks whether off-target effects of the CRISPR/Cas method could account for some of the observed phenotypes. We think this is unlikely since the *plna* R14del line was outcrossed for three generations, which should eliminate any off-target mutation not closely linked to the *plna* locus. Furthermore, all functional assays were done on large numbers of embryos that were all genotyped for the presence or absence of the *plna* R14del mutation. A functional effect caused by a

background mutation would not segregate with the *plna* R14del mutation and would not be detected.

To address this further we investigated possible off-target effects by sequencing genomic sites displaying sequence similarity with the used sgRNAs. We therefore blasted the *plna* sgRNA sequence (GGCACGGGCGCCATTCGGC) on the zebrafish genome (BLASTn for short sequences against zebrafish GRCz11), which resulted in three other regions besides the *plna* locus with significant sequence alignment (see Fig. S2 A). We designed primers to amplify these regions and sequenced these off-target regions in fish carrying the *plna* R14del mutation, but did not find any mutations in these regions. These results are presented in Figure S2.

We tried to introduce the R14del mutation in the *plnb* gene but were unsuccessful (see answer to comment 1). We did however see genetic interaction between the *plna* R14del mutation and a 5bp deletion in *plnb* on cardiac remodeling, which is another indication that this is not caused by a background mutation.

Minor points

- whilst investigating whether cardiac morphological changes occur earlier the authors say they identified one case with a similar altered tissue organisation (Figure S2). It is hard to recognise the heart - did the authors stain for any cardiac/ventricular markers?

We have indeed stained the tissue by in situ hybridization for expression of the myocardial marker, *myl7*. The *myl7* ISH results are now added to Fig.S3C,D (previous figure S2) and show the myocardium of the ventricle while the extra tissue surrounding the myocardium was negative for *myl7* expression.

- formatting of references should be checked e.g. ref 14, 23, 24, 25

We thank the reviewer for pointing to these mistakes, which are now corrected in the revised version of the manuscript.

Reviewer #4 (Remarks to the Author):

The submitted manuscript by Kamel et al. is an elegant study targeting a rare form of genetic cardiomyopathy (heterozygous phospholamban (PLN) p.Arg14del (R14del) mutation) affecting patients in the Netherlands and some other countries.

The mutation was introduced in their zebrafish model using CRISPR/Cas9 technology.

The authors detect a phenotype in the mutated fish hearts that nicely mimicks the patterns observed in patients with the mutation.

Moreover, they report found that istaroxime treatment ameliorates the in vivo Ca²⁺ dysregulation, rescues the cellular APD alternans, while it improves cardiac relaxation.

We thank the reviewer for his/her time and effort to review our work, their positive comments, and constructive remarks. Below, the points raised by the reviewer are indicated in normal font, followed by our point-by-point response in blue.

It is concluded that these effects are due to an interaction of istaroxime with PLB as reported previously in several papers. It should be acknowledged that the the displacement of PLB from

SERCA2 has been proposed as a mechanism of action but never been directly proven. The data in favour of this mechanism almost entirely derives from papers written or sponsored by the producer of that compound.

We thank the reviewer for pointing this out. We are aware that istaroxime may not only affect SERCA2 activity but also may have other activities such as inhibiting Na⁺/K⁺ ATPase (NKA) activity, which we indicated in the original version of our manuscript on page 15-16. To tone down the proposed SERCA2-inhibiting function we changed this into (line 475):

“Istaroxime is proposed to stimulate the activity of SERCA2a, thereby enhancing SR Ca²⁺ sequestration⁴⁷. In addition, it inhibits Na⁺/K⁺ ATPase (NKA) activity^{48,49}.”

It is important to note that none of the authors on this manuscript have any links to the producer of istaroxime.

A direct measurement of phospholamban or changes of its function are also lacking in the submitted manuscript. The improvement of cardiac function is shown by effects on Ca transients, contractility and APDs. One would assume decreasing diastolic Ca upon SERCA stimulation, which was not detected. This might be due to the properties of the dye used but should be discussed.

As the reviewer points out correctly, we did not observe a decrease in diastolic Ca²⁺. Diastolic Ca²⁺ concentration is not only determined by SERCA but also by diastolic sodium levels and the activity of NCX1. In zebrafish embryos, cardiac *ncx1* expression is high. Furthermore, loss of NCX1 activity was shown to result in Ca²⁺ overload of the zebrafish embryonic ventricle, while this was not observed after loss of SERCA2A activity (pmid: 16314582). Together, this indicates the complex interplay between SERCA2A and NCX1 in regulating intracellular Ca²⁺ dynamics, which we now included in our discussion on page 19 (line 569), which reads:

*“In line with our presumptions, the in vivo Ca²⁺ transient amplitude decreased in embryonic *plna-R14del* mutant fish and the Ca²⁺ reuptake duration was clearly prolonged, potentially caused by a hampered SR Ca²⁺ sequestration⁵⁹. While the diastolic Ca²⁺ concentration was unaffected in embryonic hearts of *plna R14del* mutants, it was increased in adult *plna R14del* cardiomyocytes. Importantly, in embryonic zebrafish hearts *ncx1* is strongly expressed and loss of *Ncx1* activity results in Ca²⁺ overload⁶⁰. Differences in embryonic and adult *ncx1* expression may therefore explain observed differences in diastolic Ca²⁺ levels in embryonic and adult *plna R14del* hearts. .”*

Secondly, the larger end diastolic volume seen after istaroxime seems to fit to SERCA-activating mechanisms but is also seen unchanged in the mutated fish.

Indeed, we observed a larger end-diastolic volume 30 minutes after istaroxime treatment in both the wild-type and the *plna R14del* mutant embryonic hearts (Fig. 6G). When comparing end-diastolic volumes of untreated wild type and *plna R14del* mutant we did not observe differences, which may be explained by a compensation mechanism in the *plna R14del* mutant (e.g. changes in NCX1 expression or activity, see comment above). Such a compensation mechanism may be too slow to have an effect when embryos are exposed to istaroxime for only 30 minutes.

Overall, the effects of istaroxime might also be explainable by sole Na/K-ATPase inhibition. A potential way to challenge or prove the model and hypothesis might be to perform Ca transient measurements at different and higher stimulation rates (as with the APDs). Sole Na/K-ATPase inhibitors would likely worsen diastolic function whereas additional SERCA stimulation might prevent this. A potential comparator might be cardiac glycosides.

As pointed out above we are aware of the possible dual function of istaroxime (SERCA2 stimulation and Na/K-ATPase inhibition). The suggestion by the reviewer to perform Ca^{2+} transient measurements at different and higher stimulation rates are not feasible as these measurements are conducted in whole fish larvae, which we have not been able to electrically stimulate like the single-cell preparations. Instead, to address this issue, we have compared the effects of istaroxime and the glycoside ouabain, which specifically inhibits Na/K-ATPase activity, on calcium dynamics and cardiac function in zebrafish larvae as suggested by the reviewer. As a result, we observed that istaroxime significantly enhanced stroke volume, cardiac output, and ejection fraction (Figure 6F & S9), while ouabain had no effect on these functional parameters in zebrafish larvae (shown in Fig. S11C-F). In addition, while istaroxime treatment results in a significant increase in Ca^{2+} transient amplitude, ouabain treatment had no effect on the Ca^{2+} transient amplitude (shown in Fig. S11A). This cannot be explained by ineffective ouabain treatment as we did observe a small but significant shortening of the APD using the same treatment in zebrafish larvae (shown in Fig. S12 B). From this comparison, we therefore propose that istaroxime can rescue impaired Ca^{2+} transient amplitudes in *plna* R14del mutant cardiomyocytes via interference with the PLN-SERCA interaction. As Serca stimulation is expected to result in a faster Ca^{2+} re-uptake by the SR, we now quantified the speed of the Ca^{2+} transients decay phase of *plna* R14del mutants and wild-type siblings, at baseline and after treatment with Istaroxime. We found that the decay phase is significantly slower in *plna* R14del mutants as indicated by an increased τ obtained with an exponential fit (WT: 105.74 ± 14.35 ms and *plna*-R14del: 178.6 ± 30.78 ms, $p \leq 0.05$) and that in both *plna* R14del mutants and wild-type siblings (more prominent in mutants) istaroxime clearly decreased the τ . These data are consistent with a model in which istaroxime stimulates SERCA activity and are now presented in Figure 6E.

Human data has shown no different effect of these two compounds in vitro (Wallner et al. 2017) and the clinical trial performed with istaroxime did neither show decrease in diastolic tension nor end-diastolic elastance compared to placebo but a decrease in end diastolic volume not observed in placebo (HORIZON-HF Shah 2009). The clinical program with istaroxime was stopped with two more clinical trials being withdrawn and a third one with unreported status (clinicaltrials.gov). This human data should also be discussed in the manuscript.

We thank the reviewer for pointing this out and we have therefore included an extra paragraph to the discussion on page 19 (lines 589-594) to discuss this, which reads:

*“Clinical trials with 6-24 hour istaroxime infusion in heart failure patients, however, did not demonstrate improved left ventricular diastolic volumes^{63,64}, which is expected from improved SR Ca^{2+} reuptake. This may be explained by the Na/K-ATPase inhibiting activity of istaroxime, which leads to increased diastolic Ca^{2+} . The *plna* R14del zebrafish model may help to screen for more selective compounds that only improve SERCA2a activity without affecting Na/K-ATPase activity to benefit PLN R14del carriers. “*

REVIEWER COMMENTS

Reviewer #1 (Remarks to the Author):

The authors have improved the manuscript, but there are still some concerns. A main conclusion is that Ca²⁺ dysregulation is associated with PLN R14del, which can be rescued with istaroxime. The authors make some progress in this regard, but given the importance of this conclusion, the data in support need to be solid. However, this is not yet the case for reduced Ca²⁺ uptake, which many of the readouts focus on. For example, the authors use APD alternans as a surrogate for reduced Ca²⁺ uptake. However, APD alternans can occur independent of reduced Ca²⁺ uptake. The best result would be to show reduced Ca²⁺ uptake results that are direct and convincing. For example, Ca²⁺ recovery, Ca²⁺ duration, and Ca²⁺ decay (τ) should all be measurable from the Ca²⁺ transient and consistent since they are all related to Ca²⁺ uptake. However, the data do not show this. Figure 5 (top, adult) only shows duration and decay and the n is very small (4-5). Figure 5 (bottom, embryo) only shows recovery and decay and they don't agree. Furthermore, the fact that istaroxime increases uptake is not a surprise since it activates SERCA2a. Finally, if Ca²⁺ uptake was slower in PLN R14del, then Ca²⁺ transient alternans (and contraction alternans) should be obvious; however, this was not observed. In sum, the data are not convincing enough to say that Ca²⁺ uptake is slower with pIna R14del.

The pulse alternans are not obvious, and the analysis provided in the rebuttal (figure R1.2) agrees. Accordingly, the authors cannot say that pulse alternans is shown (lines 408-410). Increased variability is about all that can be said.

Reviewer #3 (Remarks to the Author):

The authors have addressed all my concerns.

Reviewer #4 (Remarks to the Author):

Comprehensive data and additional discussion was added with particular focus on Ca handling and contractility by the authors. The manuscript further improved considerably. There are no more issues.

Reviewer #5 (Remarks to the Author):

The authors have done an impressive amount of additional work in response to the reviewers' comments. Many of the issues raised have been adequately addressed. However, the issues of zebrafish PLN paralogs (PLNa and PLNb) and orthologs (e.g. sarcolipin) are common to any ZF study because they are not sufficiently understood. The concept of the whole genome duplication in the teleost lineage after divergence from the primate lineage should be addressed as a limitation of the study. The discussion of the mechanisms of action of istaroxime has been improved by the additional studies but not fully resolved and should also be considered as somewhat of a limitation.

Reviewer #1 (Remarks to the Author):

The authors have improved the manuscript, but there are still some concerns. A main conclusion is that Ca²⁺ dysregulation is associated with PLN R14del, which can be rescued with istaroxime. The authors make some progress in this regard, but given the importance of this conclusion, the data in support need to be solid. However, this is not yet the case for reduced Ca²⁺ uptake, which many of the readouts focus on.

First of all, we thank the reviewer for his/her time and efforts to review our manuscript. We took all comments of this reviewer to heart and made changes to the manuscript accordingly. Our responses to the reviewer's comments are given in a point-by-point fashion below, repeating each of the reviewer's comments in bold, directly followed by our response. Changes made to the manuscript appear in a distinct colour in the revised manuscript.

For example, the authors use APD alternans as a surrogate for reduced Ca²⁺ uptake. However, APD alternans can occur independent of reduced Ca²⁺ uptake.

It was not our intention to present APD alternans as a surrogate for reduced Ca²⁺ uptake, and have not done so anywhere in the manuscript. We fully agree that APD alternans can have various mechanisms, as nicely reviewed by Kulkarni et al. (Kulkarni et al., 2019; PMID: 31617437). We think that the APD alternans observed at 4 Hz in the present study (Figure 4D) is related to altered Ca²⁺ homeostasis rather than changes in ion channel function for the following reasons. The *average* APD does not differ between WT and plna R14del myocytes in all frequencies tested (Figure 4C), which is supported by the virtually overlapping of major ion currents underlying in the cardiac action potential (Figure S6). The absence of average APD differences between wild-type and plna R14del cardiomyocytes thus strongly argues against a membrane voltage or action potential (AP)–driven mechanisms (Kulkarni et al., 2019) of our observed APD alternans at 4 Hz. This is further supported by the lack of differences in the ion currents we studied. In contrast, we did find Ca²⁺ abnormalities in PLN R14del cells, which is suggested as a cause of alternans (Kihara & Morgan, 1991, PMID: 1750531; Kulkarni et al., 2019, PMID: 31617437). We therefore think that the APD alternans is due to Ca²⁺ abnormalities rather than other ion channel related mechanisms.

We have adapted the discussion section line 558-571 to discuss this in more detail which now reads as: *“In accordance with our echocardiography measurements, we noticed a clear alternans between consecutive APs in plna R14del cardiomyocytes during patch clamp experiments at the high pacing frequency of 4 Hz. Alternans may be a membrane voltage or action potential (AP)–driven phenomenon or due to Ca²⁺,^{45,58}. In the present study, we found no differences in average APD in a wide range of frequencies between WT and plna R14del myocytes, which is supported by the virtually overlapping of major ion currents underlying in the cardiac action potential. This strongly argues against a membrane voltage or action potential (AP)–driven mechanisms of our observed APD alternans⁴⁵. A plausible mechanism underlying the alternans is a decreased pump rate of SR Ca²⁺ pumps, resulting in alternating strong and weak beats^{58,59}. Also, concomitant slow transportation of Ca²⁺ from the uptake compartment to the release compartment in the SR has been suggested as a cause of alternans⁵⁸. Corroborating such a model, we observed a slower Ca²⁺ reuptake in plna R14del cardiomyocytes and by application of istaroxime we were able to restore reduced Ca²⁺ amplitudes and decay phase, and prevent electrical alternans. Istaroxime is a drug that acts by releasing PLN from SERCA and thereby stimulates SR Ca²⁺ loading and release⁴⁸.”*

The best result would be to show reduced Ca²⁺ uptake results that are direct and convincing. For example, Ca²⁺ recovery, Ca²⁺ duration, and Ca²⁺ decay (tau) should all be measurable from the Ca²⁺ transient and consistent since they are all related to Ca²⁺ uptake. However, the data do not show this. Figure 5 (top, adult) only shows duration and decay and the n is very small (4-5). Figure 5 (bottom, embryo) only shows recovery and decay and they don't agree. Furthermore, the fact that istaroxime increases uptake is not a surprise since it activates SERCA2a.

We respectfully disagree with this reviewer. The parameters Ca²⁺ recovery, duration and decay are indeed all derived from the same calcium transients recorded, and therefore may be expected to reflect the phenotype in a consistent manner. However, some parameters are more sensitive to noise in the recordings, especially the recovery and duration data that depend on the identification of single points on the calcium transient where e.g. 90% of reuptake is achieved. The decay parameter (tau) is more robust in that respect, as it is determined by fitting all time points of the reuptake phase of the transient with an exponential function, thereby maximising the use of information in the recording. This makes the decay parameter the most robust representation of the Ca²⁺ reuptake speed in our study.

We have performed ratiometric Ca²⁺ measurements in isolated adult cardiomyocytes with Indo-1 and calculated intracellular [Ca²⁺]_i. We found a significantly increased diastolic [Ca²⁺]_i, longer Ca²⁺ transients at 20% and 50% of the transient, and slower decay of the Ca²⁺ transient, which all are consistent with a slower SR Ca²⁺ reuptake. These data are included as text in the manuscript at line 446-451 and are shown in Figure 5 B, C, E and F. In 3-day old embryos, we have performed *in vivo* analysis of intracellular Ca²⁺ dynamics due to their optical transparency. We found a Ca²⁺ transient amplitude which was on average 26% lower in *plna* R14del mutant embryos compared to wild-type siblings (Figure 5J). In addition, we observed a trend towards prolonged Ca²⁺ transient recovery time and importantly, a significantly slower decay of the Ca²⁺ transient in *plna* R14del mutants compared to WT (Figure 5 I, L).

Our results thus clearly demonstrate that intracellular Ca²⁺ dynamics in cardiomyocytes of the embryonic and adult heart are affected by the *plna* R14del mutation and indicate that *plna* R14del impairs SR Ca²⁺ reuptake.

We agree that not all measured Ca²⁺ parameters in embryonic and adult heart match completely. For example, recovery time shows a tendency to prolongation in embryonic heart, but the Tau of the Ca²⁺ transient decay was significantly prolonged. We think that it is related to the rather noisy Ca²⁺ transient signals making determination of the recovery time more difficult, as explained above. Methodological differences between calcium transient recordings in adult and embryonic cardiomyocytes/hearts have to be taken into account as well. Calcium transient duration parameters have less added value in embryonic hearts, as the upstroke time is fairly slow in these hearts and the frequency of transients is not standardized by pacing. This could be optimized in adult fish by recording calcium transients on isolated cardiomyocytes, making use of a ratiometric dye. In both cases, the tau of decay is the most robust measure of SR Ca²⁺ reuptake, as described above, which made us decide to focus on that parameter. In addition, we found an increased absolute diastolic [Ca²⁺]_i in adult cells and unaltered normalized relative diastolic Ca²⁺ levels in embryonic heart of *plna* R14del, but such discrepancy is acknowledged and discussed at line 578-584 of the manuscript. The measures of diastolic [Ca²⁺]_i using Indo-1 in adult cardiomyocytes and GCaMP6f in embryos are not fully comparable, as the calibrated Indo-1 data gives an absolute concentration, whereas the GCaMP6f *in vivo* signal cannot be calibrated, which we now mention in the discussion at line 577-580.

The data presented in Figure 5 resulted from 3 biological repeats, which gave consistent results and demonstrated a significant difference between groups. We spent a lot of time on the cardiomyocyte isolations from the adult zebrafish heart for the ratiometric Ca²⁺ measurements and

increasing the n would most likely not change the outcome of these data rather than significantly delaying the publication of these results.

Finally, if Ca²⁺ uptake was slower in PLN R14del, then Ca²⁺ transient alternans (and contraction alternans) should be obvious; however, this was not observed.

We apologize for being unclear, but there is a technical explanation for the reviewer suggested contradiction. In adult myocytes, APD alternans was observed at 4 Hz stimulation using patch clamp methodology and not at slower pacing frequencies. Unfortunately, we were not able to reach a 4 Hz stimulation with the use of field stimulation during the Indo-1 Ca²⁺ measurements in adult myocytes. Consequently, we did not observe any alternans with the Ca²⁺ transients in adult myocytes. We have included this information at line 571-573 in the revised MS, which reads: *'We did not find Ca²⁺ transient alternans, possibly because we were not able to reach the required 4 Hz stimulation with the use of field stimulation during the Indo-1 in adult myocytes.'*

In sum, the data are not convincing enough to say that Ca²⁺ uptake is slower with plna R14del.

We respectfully disagree with the reviewer. The increased tau of the Ca²⁺ transients in both embryonic hearts and isolated adult cardiomyocytes are indicative for slower Ca²⁺ uptake, as is the increased diastolic Ca²⁺ observed in the isolate adult cardiomyocytes.

The pulse alternans are not obvious, and the analysis provided in the rebuttal (figure R1.2) agrees. Accordingly, the authors cannot say that pulse alternans is shown (lines 408-410). Increased variability is about all that can be said.

Based on the reviewer's suggestion we have replaced 'pulse alternans' with 'variations in outflow peak velocities' when describing the echocardiography results in lines 407-410 and shown in Figure 3.

Reviewer #3 (Remarks to the Author):

The authors have addressed all my concerns.

We thank the reviewer for his/her time and efforts to review our manuscript.

Reviewer #4 (Remarks to the Author):

Comprehensive data and additional discussion was added with particular focus on Ca handling and contractility by the authors. The manuscript further improved considerably. There are no more issues.

We thank the reviewer for his/her time and efforts to review our manuscript.

Reviewer #5 (Remarks to the Author):

The authors have done an impressive amount of additional work in response to the reviewers' comments. Many of the issues raised have been adequately addressed. However, the issues of zebrafish PLN paralogs (PLNa and PLNb) and orthologs (e.g. sarcolipin) are common to any ZF study because they are not sufficiently understood. The concept of the whole genome duplication in the teleost lineage after divergence from the primate lineage should be addressed as a

limitation of the study. The discussion of the mechanisms of action of istaroxime has been improved by the additional studies but not fully resolved and should also be considered as somewhat of a limitation.

we thank the reviewer for his/her time and efforts to review our manuscript. We agree that some limitations of our study should be highlighted which we did by adding an extra paragraph to the discussion on page 20 (lines 619-626):*'Some limitations of this study should be considered. First, the pln gene has been duplicated in the zebrafish due to the teleost-specific genome^{67,68}. Duplicated genes can act redundant or may diverge in their function, which may complicate interpolations to mammalian models. As both plna and plnb are highly expressed in the ventricle and our studies show their genetic interaction, we favor the hypothesis that plna and plnb have redundant roles in the zebrafish ventricle. A second limitation is that istaroxime used in this study has a dual function as it stimulates SERCA2 activity and inhibits Na⁺/K⁺ ATPase (NKA) activity^{49,50}. The plna R14del zebrafish model may help to screen for more selective compounds that only improve SERCA2a activity without affecting Na/K-ATPase activity to benefit PLN R14del carriers.'*

Changes made to the manuscript appear in a distinct colour in the revised manuscript.

REVIEWERS' COMMENTS

Reviewer #1 (Remarks to the Author):

The authors have addressed my concerns and I have no further comments.